# Twenty Years of Research on Cyclodextrin Conjugates with PAMAM Dendrimers

**DOI:** 10.3390/pharmaceutics13050697

**Published:** 2021-05-11

**Authors:** Hidetoshi Arima

**Affiliations:** School of Pharmacy, Daiichi University of Pharmacy, Fukuoka 815-8511, Japan; h-arima@daiichi-cps.ac.jp; Tel.: +81-92-541-0161

**Keywords:** cyclodextrin, polyamidoamine, dendrimer, conjugate, gene, oligonucleotide, polypseudorotaxane, antitumor drugs, anti-inflammatory effect, anti-amyloid effect

## Abstract

Recently, the number of gene and oligonucleotide drugs are increasing. Of various drug delivery systems (DDSs) for gene and oligonucleotide drugs, few examples of the clinical application of polymer as drug carriers are known, despite development of the novel polymers has been progressing. Cyclodextrin (CD) conjugates with starburst polyamidoamine (PAMAM) dendrimer (CDEs), as a new type of polymer-based carriers, were first published in 2001. After that, galactose-, lactose-, mannose-, fucose-, folate-, and polyethyleneglycol (PEG)-appended CDEs have been prepared for passive and active targeting for gene, oligonucleotide, and low-molecular-weight drugs. PEG-appended CDE formed polypsuedorotaxanes with α-CD and γ-CD, which are useful for a sustained release system of gene and oligonucleotide drugs. Interestingly, CDEs were found to have anti-inflammatory effects and anti-amyloid effects themselves, which have potential as active pharmaceutical ingredients. Most recently, CDE is reported to be a useful Cas9-RNA ribonucleoproteins (Cas9 RNP) carrier that induces genome editing in the neuron and brain. In this review, the history and progression of CDEs are overviewed.

## 1. Introduction

The modalities of pharmaceuticals are diversifying, and genes, oligonucleotides, cells, and digital medicines have been developed recently [1]. In particular, regulatory approvals for gene drugs, oligonucleotide drugs, and regenerative medicine products have recently come one after another. To date, approximately 20 cellular and gene therapy products have been approved by the regulatory authorities in the world, especially, in the United States, Europe, and/or Japan. One of them, Zolgensma^®^ (onasemnogene abeparvovec-xioi), using an adeno-associated virus vector, is now on the market. In addition, Colategene^®^, a plasmid DNA preparation without a DNA carrier, was approved in 2019 in Japan. It should be noted that tumor lysing virus products are also on the market. Meanwhile, a total of 12 oligonucleotide drugs including 8 antisense drugs, 1 aptamer drug, 1 CpG-oligo drug, and 2 siRNA drugs have been approved by the US Food and Drug Administration (FDA), EU European Medicines Agency (EMA), and/or Japan Pharmaceuticals and Medical Devices Agency (PMDA) [2]. It is worth noting that the BNT162b2, which is an mRNA vaccine encoding a P2 mutant spike protein of COVID-19 and formulated as an RNA-lipid nanoparticle (LNP) of nucleoside-modified mRNA was approved in 2020 [3]. Hence, the development of gene drugs and mRNA vaccines is progressing.

In general, of oligonucleotide drugs, antisense drugs (molecular weight of about 7000) are not required to be equipped with drug delivery system (DDS) technology, because various chemical modifications of their molecules improve enzyme stability and hybridization ability with target molecules, as well as cell uptake, which is sufficient for the expression of therapeutic effects, although the efficiency is still low. Meanwhile, siRNA (molecular weight of about 14,000), which is double-stranded RNA, has low cell membrane permeability due to its high molecular weight and negative surface charge, and excessive chemical modification of siRNA molecules reduces the RNAi effect [4]. Hence, DDS technology has been utilized for siRNA drugs. In fact, two siRNA preparations developed by Alnylam Pharmaceuticals have been used DDS technology. That is, Onpattoro^®^ (sodium patisiran) is an LNP formulation employing DDS technology (Stable Nucleic Acid Lipid Nanoparticle; SNALP) introduced by Tekmira pharmaceuticals (now Arbutus Biopharma) [5]. The components of LNP are 65.0 mg DLin-MC3-DMA (pH-responsive cationic lipids for endosomal escape), 8.0 mg PEG2000-C-DMG (passive targeting), 16.5 mg DSPC (LNP membrane component), and 31.0 mg cholesterol (LNP membrane component). The interesting point regarding DDS is that the LNP itself does not have the ability to target the liver, but it is an active targeting agent that acquires the ability to target liver parenchymal cells by adsorbing endogenous ApoE after intravenous administration [5]. Another DDS technology used in Givlaari^®^ (givosiran) is a chemically modified siRNA drug conjugation with three N-acetylgalactosamine (GalNAc) molecules for targeting to asialoglycoprotein receptor (ASGP-R), which is highly expressed on liver parenchymal cell membranes [6].

As described above, both Onpattro^®^ and Givlaari^®^ are siRNA products using DDS technology that targets liver parenchymal cells. Since the target diseases of siRNA preparations naturally extend to organs and cells throughout the body, other DDS technologies are also being developed [7]. Alnylam Pharmaceuticals has developed a systemic siRNA preparation for Alzheimer’s disease, Huntington’s disease, and amyotrophic lateral sclerosis via intrathecal injection for a wide range of central nervous system (CNS) diseases. In addition, Ionis Pharmaceuticals, a pioneer in the development of antisense oligonucleotide drugs, has introduced GalNAc’s DDS technology, and mutual collaboration on active targeting of antisense oligonucleotides is being implemented. Moreover, examples of the use of unit polyion complexes and exosomes as oligonucleotide drug carriers, in addition to aptamers and siRNA conjugates with antibodies, are being investigated, and we look forward to good results in future research and clinical trials.

## 2. Cyclodextrins Conjugates with PAMAM Dendrimers for Gene and Oligonucleotide Drug Delivery 

Polymeric DDS technology has achieved great development in the last two decades [8]. There are many excellent reviews regarding the use of polymers for DDS for genes and oligonucleotide drugs [9]. However, few examples of the clinical application of polymer as drug carriers are known, despite development of the novel polymers has been progressing. Therefore, the development of novel and practical polymer-based carriers for these drugs has been attempted. Cyclodextrins (CDs) are cyclic -1,4-linked oligosaccharides of D-glucopyranose containing a hydrophobic central cavity and hydrophilic outer surface [10,11]. The structure and properties are shown in Figure 1 and Table 1, respectively. The most common CDs are α-, β-, and γ-CDs, which consist of six, seven, and eight D-glucopyranose units, respectively, although new CDs consisting of three and four D-glucopyranose units was synthesized by Yamada et al. [12]. In addition, Watanabe et al. reported a cyclomaltopentaose cyclized by an α-1,6-linkage from starch [13]. Moreover, Saenger et al. reported more than large-ring CDs composed of more than nine glucopyranose units [14]. CDs are known to form inclusion complexes with a variety of guest molecules in solution and in a solid state. A number of hydrophilic CD derivatives have been developing [15]. Among them, 2-hydroxypropyl-β-CD (HP-β-CD) and sulfobutylether-β-CD (SBE-β-CD) are the most common, since drugs containing HP-β-CD and SBE-β-CD have been commercialized in the USA, EU, and Japan [16]. More than 50 CD-containing products have been used in the world. It should be noted that FDA approved Remdesivir^®^ containing the excipient

SBE-β-CD, and FDA issued emergency use authorization the Janssen COVID-19 Vaccine containing HP-β-CD for treatment and prevention of COVID-19, respectively. Meanwhile, in Japan, branched CDs such as glucosyl-β-CD (G_1_-β-CD), maltosyl-β-CD (G_2_-β-CD), and glucuronylglucosyl-β-CD (GUG-β-CD) can be available. In particular, of branched CDs, GUG-β-CD is composed of glucose and glucuronic acid (Figure 1 and Table 1) and prepared by the oxidation of maltosyl-β-CD with *Pseudogluconobacter saccharoketogenes* [17]. GUG-β-CD has lower hemolytic activity and cytotoxicity than β-CD and G_2_-β-CD [18]. In addition, GUG-β-CD showed greater affinity for the basic drugs, compared with β-CD and G_2_-β-CD, due to electrostatic interaction of its carboxylate anion with a positive charge of basic drugs. Thus, GUG-β-CD may be useful as a safe solubilizing agent, particularly for basic drugs [15]. Moreover, GUG-β-CD has an advantage for easy conjugation of a primary amino group with a carboxyl group in the molecule [18]. Intriguingly, GUG-β-CD is acknowledged to inhibit misfolding of the transthyretin (TTR), a β-sheet-rich protein, which, in turn, suppresses TTR amyloid formation [19]. Meanwhile, the dissolution of lipids from biomembranes through complexation with CDs, especially methyl-β-CD and HP-β-CD, may be useful in the study of cellular biology, e.g., caveolae, lipid rafts, and cholesterol transport. It is not certain that CDs are not polymers, but it is acknowledged that CD-containing polymers and CD conjugates with cationic polymers such as polyethyleneimine (PEI) and starburst polyamidoamine (PAMAM) dendrimers (PAMAM dendrimers), and CD-based supramolecules such as polyrotaxanes (PRX) and polypseudorotaxanes (PPRX) with linear polymers have been utilizing as carriers for delivery for gene and oligonucleotide drugs [20,21,22,23]. Additionally, CD-based supramolecular nanoassemblies that are sensitive to chemical, biological, and physical stimuli have been designed and developed [24]. Among them, Dr. Davis et al. developed a nanoassembly to conjugate a neutral stabilizing polymer, polyethylene glycol (PEG), to a hydrophobic small molecule and adamantane (AD), which forms strong inclusion complexes with β-CD [25,26]. In this manner, nanoparticles could be noncovalently stabilized, and this approach was extended to allow the incorporation of targeting ligands via the preparation of AD-PEG-ligand conjugates. Actually, the RNAi/oligonucleotide nanoparticle delivery (RONDEL) system such as the CALAA-01 drug product developed by Calando Pharmaceuticals was terminated.

Of polymeric DDS materials, dendrimers are known to be a unique polymer because of having nanosized, apparently symmetric molecules with a well-defined, homogeneous, and monodisperse structure that has a normally NH_3_ or ethylenediamine core, an inner shell, and an outer shell [27]. Of dendrimers, PAMAM dendrimers are one of the most popular polymers used as a nonviral gene carrier, because they have positively charged primary amine groups on the peripheral ends to allow interaction with the negatively charged gene and oligonucleotide drugs [28]. Additionally, PAMAM dendrimers have the potential to overcome several physiological barriers to achieve effective transfection, namely, it is known that polyplexes could be internalized into cells by endocytosis-forming endosomes [29]. These endosomes progress into an endolysosome state, which has an acidic environment to digest external materials. Therefore, most of the complexes in endolysosomes will collapse. However, PAMAM dendrimers possess the proton buffering ability in their tertiary amine (hypothesized proton sponge effect) to be needed to overcome this acidic environment of the endolysosomes [30]. However, the transfer activity of PAMAM dendrimers for gene and oligonucleotide drugs is insufficient, and cytotoxic activity of PAMAM dendrimers with high generation requires great caution and careful attention [31]. Therefore, the drawback of PAMAM dendrimers should be improved through the insertion of additional functional groups.

Many studies have investigated the enhanced transfection efficiency of modified PAMAM dendrimer derivatives [32]. Grafting of specific ligands for certain receptors (e.g., RGD sequence for α_v_β_3_ integrins) provides enhanced targeting effects to carriers. Additionally, an arginine-conjugated PAMAM dendrimer shows enhanced transfection efficiency, compared to the native PAMAM dendrimer [33]. Moreover, the acetylation of PAMAM dendrimers is known to decrease cytotoxicity while maintaining membrane permeability [34]. Additionally, PEG-appended (PEGylated) PAMAM dendrimers are known to enhance efficacy and mitigate toxicity [35].

In the review, the potential of various multifunctional CD/PAMAM dendrimer conjugates (CDEs) is summarized. 

## 3. Parent CD/PAMAM Dendrimer Conjugates (CDE) for Delivery Gene and Oligonucleotide Drugs

### 3.1. CDE for pDNA Delivery

Thus far, Hirayama et al. reported the potential uses of CD conjugates with biphenylyl acetic acid and prednisolone for colon-specific drug delivery have been reported [36]. Meanwhile, Arima et al. [37] synthesized the PAMAM dendrimer (G2) conjugates with α-CD, β-CD, and γ-CD (α-CDE, β-CDE, γ-CDE, respectively) at 1:1 of average degrees of substitution (DS) of CD (DSC) in anticipation of the following synergic effect, i.e., (1) PAMAM dendrimer has the abilities to complex with pDNA and oligonucleotides and to enhance their cellular uptake and (2) CDs have a disrupting effect on biological membranes by the complexation with membrane constituents such as phospholipids and cholesterols. Herein, DSC represents the number of CD units bound to a PAMAM dendrimer molecule, although the DS is used in CD chemistry for describing the number of substituents on CD molecules. To prepare α-CDE and β-CDE, parent α-CD and β-CD were first mono-6-*O*-tolylated, while to prepare γ-CDE, γ-CD was first mono-6-*O*-naphthalenesulfonylated, indicating that these synthesis steps seem to be slightly unwieldy (Figure 2). Using these CDEs, Arima et al. investigated the effects of three CDEs (G2) on the gene transfer efficiency in the cells, and the enhancing effects were compared with commercial transfection reagents, Lipofectin^®^, and TransFast^®^ [37]. Each CDE formed the complexes with pDNA and protected from enzymatic degradation of pDNA by DNase I. CDEs (G2, DSC1) showed a potent reporter (luciferase) gene expression. Among three CDEs (G2, DSC1), α-CDE (G2, DSC1) showed the highest gene transfer activity—approximately 100 times higher than those of PAMAM dendrimer (G2, DSC1) alone and of the physical mixture of PAMAM dendrimer (G2) and α-CD in NIH3T3 and RAW264.7 cells were observed, indicating that conjugation of PAMAM dendrimer (G2) with α-CD would be crucial for enhancing gene transfer activity. In addition, the gene transfer activity of α-CDE (G2, DSC1) was superior to that of Lipofectin^®^. The enhancing gene transfer effect of α-CDE (G2, DSC1) may be attributable to increasing the cellular association and changing the intracellular trafficking of pDNA. Additionally, it is interesting that free α-CDE (G2, DSC1) also contributed to the enhancing effects of its gene transfer activity. However, it is still unknown the detailed mechanism by which α-CDE (G2, DS1) has higher gene transfer activity than β-CDE (G2, DSC1) and γ-CDE (G2, DSC1) under experimental conditions. In addition, as described below, PAMAM dendrimer conjugates with GUG-β-CD (G2) showed transfection activity higher than α-CDE (G2, DSC1); therefore, a further detailed study is needed to clarify the mechanism by which CDEs enhance gene transfer activity. Importantly, α-CDE (G2, DSC1) showed cytotoxicity only very slightly. Taken together, these findings suggest that α-CDE could be a new preferable nonviral vector of pDNA [37].

Next, the effects of the generation of PAMAM dendrimers in α-CDE on gene transfer activity were investigated [38]. The gene transfer activity of α-CDEs (G2, G3, and G4, DSC1) was higher than that of the corresponding PAMAM dendrimer alone in NIH3T3 and RAW264.7 cells. Of three CDEs (G2, G3, and G4, DS1), α-CDE (G3, DSC1) had the highest superior gene transfer activity, which was comparable to that of TransFast^®^ in NIH3T3 cells. Remarkably, α-CDE (G3, DSC1) was suggested to improve endosomal escape after entering cells through the synergetic effects of the proton sponge effect of PAMAM dendrimer (G3) and interaction of α-CD in an α-CDE (G3, DSC1) molecule with biomembrane constituents such as phospholipids, possibly in endolysosomes after transfection. Although the detailed mechanism by which α-CDE (G3, DSC1) has superior gene transfer activity to α-CDE (G4, DSC1), the number of the primary amino group of the former is twice higher than that of the latter, hence suggesting that the proton sponge effect of the former should be higher than that of the latter. Cytotoxicity, as one of the possible reasons, might be involved in the unpredicted results, and therefore, a further elaborative study is necessary. Collectively, these results suggest that α-CDEs (G2, G3, and G4, DSC1), particularly α-CDE (G3, DSC1) could be novel nonviral gene transfer agents [38].

In order to optimize the chemical structure of α-CDE (G3) as a nonviral vector, α-CDE (G3) with various DSC values of 1.1, 2.4, and 5.4 were prepared [36]. The membrane-disruptive ability of α-CDE (G3) on liposomes encapsulating calcein, an artificial model of endolysosomes, and their cytotoxicity to NIH3T3 and HepG2 cells increased with an increase in the DSC value. In vitro gene transfer activity of α-CDE (G3, DSC2.4) in both NIH3T3 and HepG2 cells augmented as the charge (N/P) ratio increased, and the activity of α-CDE (G3, DSC2.4) was the highest at higher charge (N/P) ratios among PAMAM dendrimer (G3, DSC2.4), the three α-CDEs (G3), and TransFast^®^. After intravenous administration of pDNA complexes in mice, α-CDE (G3, DSC2.4) delivered pDNA more efficiently in the spleen, liver, and kidney, compared with PAMAM dendrimer and other α-CDE (G3, DSC1.1 and 5.4). These results in the studies using the α-CDE systems indicate that there is an optimal point regarding a balance between the number of a primary amino group and α-CD. Furthermore, it should be emphasized that cytotoxicity of PAMAM dendrimers was strikingly suppressed by conjugation of CD, probably owing to a steric hindrance and decrease in the positive ζ-potential value of PAMAM dendrimer and submicron particles with pDNA. In conclusion, the potential use of α-CDE (G3, DSC2.4) could be expected as a nonviral vector in vitro and in vivo, and these data may be useful for the design of α-CDEs with other nonviral vectors [36]. Figure 3 shows a result of optimization of α-CDEs as a pDNA carrier.

### 3.2. CDE for Short Hairpin RNA Expressing pDNA (shpDNA) Delivery

Short hairpin RNA (shRNA) is an alternative way to prepare small interfering RNA (siRNA) sequences for delivery to cells that can be expressed in situ from pDNA or from virus-derived constructs. A potentially beneficial effect of in situ shRNA expression may be to ensure that RNA interference (RNAi) effects are more sustained (months) than might usually be expected postdelivery of synthetic siRNAs (few weeks maximum). For these reasons, shRNA is considered a useful RNAi therapeutic agent with future potential [38,39,40].

Tsutsumi et al. evaluated the potential of α-CDE (G3, DSC2.4) as a novel carrier of shRNA expressing pDNA (shpDNA), because it showed the highest gene transfer activity among various α-CDEs [41]. The shpDNA transfer activity of α-CDE (G3, DSC2.4) was compared with that of PAMAM dendrimer (G3). α-CDE (G3, DSC2.4) formed a stable and condensed complex with shpDNA and induced a conformational transition of shpDNA in solution even in the low charge (N/P) ratios. In addition, α-CDE (G3, DSC2.4) evidently repressed the enzymatic degradation of shpDNA by DNase I. The shpDNA complex with α-CDE (G3, DSC2.4) at a charge (N/P) ratio of 20/1 provoked the most potent RNAi effects in cells transiently and stably expressing the GL3 and GL2 luciferase genes without cytotoxicity among the complexes with the various charge (N/P) ratios. Additionally, the RNAi effects were outstandingly enhanced by the further addition of adequate amounts of siRNA to the shpDNA complex with α-CDE (G3, DSC2.4). Collectively, the prominent RNAi effects of the shpDNA complex with α-CDE (G3, DSC2.4) could be ascribed to the stabilizing effect of α-CDE (G3, DS2.4) on enzymatic degradation of shpDNA and negligible cytotoxicity. These results suggest that α-CDE (G3, DSC2.4) possesses the potential to be a novel carrier for shpDNA and siRNA [41].

### 3.3. CDE for siRNA Delivery

As described above, RNAi is a sequence-specific, gene-silencing mechanism triggered by double-stranded RNA and powerful tools for a gene function study and RNAi therapy. Although siRNAs offer several advantages as potential new drugs to treat various diseases, the efficient delivery system of siRNAs in vivo remains a crucial challenge for achieving the desired RNAi effect in clinical use. Recently, FDA, EMA, and PMDA approval of siRNA therapeutics garnered a new hope in siRNA therapeutics. However, the development of the novel siRNA carrier system should be needed. As the first step toward an evaluation of the potential use of the α-CDE (G3, DSC2.4) for siRNA carrier, the ternary complexes of α-CDE (G3, DSC2.4) or the transfection reagents such as Lipofactamine^®^2000, TransFast^®^, and Lipofectin^®^ with pDNA and siRNA were prepared, and their RNAi effects, cytotoxicity, physicochemical properties, and intracellular distribution were compared [39]. Herein, the pGL2 control vector (pGL2) and pGL3 control vector (pGL3) encoding the firefly luciferase gene and the two corresponding siRNAs (siGL2 and siGL3) were used. In addition, it is well known that PAMAM dendrimers have distinct advantages, such as high drug-loading capacity at the surface terminal for conjugation or interior space for encapsulation, depending on the generation of PAMAM dendrimers and a charge (N/P) ratio of pDNA or oligonucleotides and PAMAM dendrimers. In addition, PAMAM dendrimers with adequate DS values of CD, PEG, and/or targeting ligand provide similarly high drug-loading capacity. The ternary complexes of pGL3/siGL3/α-CDE (G3, DSC2.4) showed the potent RNAi effects with negligible cytotoxicity, compared to those of the transfection reagents in various cells. α-CDE (G3, DSC2.4) strongly interacted with both pDNA and siRNA and inhibited siRNA degradation by serum, compared to those of the transfection reagents. α-CDE (G3, DSC2.4) allowed fluorescent-labeled siRNA to distribute in the cytoplasm, whereas the transfection reagents resided in both nucleus and cytoplasm in NIH3T3 cells. Moreover, the binary complex of siRNA/α-CDE (G3, DSC2.4) elicited the noteworthy RNAi effect in NIH3T3 cells transiently and stably expressing luciferase gene. These results suggest that α-CDE (G3, DSC2.4) may be utilized as a novel carrier for siRNA [42].

Considering siRNA therapeutics, Arima et al. next prepared various binary siRNA complexes with α-CDE (G3, DSC2.4) and examined the physicochemical properties, serum resistance, in vitro RNAi effects on endogenous gene expression, cytotoxicity, interferon response, hemolytic activity, cellular association, and intracellular distribution [43]. α-CDE (G3, DSC2.4) interacted with siRNA and repressed siRNA degradation by serum. The siRNA complex with α-CDE (G3, DSC2.4) showed the potent RNAi effects against Lamin A/C and Fas expression with negligible cytotoxicity and hemolytic activity, compared to those of the transfection reagents in Colon-26-luc cells and NIH3T3-luc cells, which are expressing luciferase. Cell-death patterns induced by siRNA polyplexes with α-CDE (G3, DSC2.4) and a linear PEI were different from siRNA lipoplexes with Lipofectamine^®^2000 and RNAiFect^®^. α-CDE (G3, DSC2.4) delivered fluorescent-labeled siRNA to the cytoplasm, not the nucleus, after transfection in NIH3T3-luc cells. Taken together, α-CDE (G3, DSC2.4) could be potentially used as a siRNA carrier to provide the RNAi effect on endogenous gene expression with negligible cytotoxicity [43].

### 3.4. α-CDE (G3, DSC2) for the Other Negatively Charged Drug Delivery

Despite the development of antiretroviral therapy against the human immunodeficiency virus (HIV), suppression of the virus from the body remains in progress. A “kick and kill” strategy proposes a “kick” of the latent HIV to an active HIV to eventually be “killed.” Latency-reverting agents that can perform the “kick” function are under development and have shown promise. Tateishi et al. developed L-heptanolyphosphatidyl inositol pentakisphosphate (L-HIPRO), an artificial phosphoinositide as an antagonist of PIP2 to suppress the membrane localization of Pr55^Gag^ [44]. However, L-HIPRO has too dense a negative phosphate charge to enter cells. To enter L-HIPRO, α-CDE (G3, DSC2) was used as a carrier for L-HIPRO. The L-HIPRO complex with α-CDE (G3, DSC2) suppressed membrane localization of Pr55^Gag^ and subsequent HIV virus release and induced apoptosis of the host cells. Thus, the complex encloses the penetrated virus components, locks them in the host cells, and destroys the virus by means of apopmechantosis of the host cells. These results appear to provide a promising step toward the goal of HIV eradication from the body [41]. Figure 4 summarizes its outstanding features of α-CDE (G3).

## 4. GUG-β-CD/Dendrimer Conjugates (GUG-β-CDE) as pDNA and siRNA Delivery

### 4.1. GUG-β-CDE for pDNA Delivery

As described above, GUG-β-CD has a carboxyl group, indicating an easy conjugation with an amino group of PAMAM dendrimer [15], compared with α-CDE and β-CDE. Figure 5 shows the chemical structures of GUG-β-CDE (G2, DSC1) and GUG-β-CDE (G3, DSC1).

Herein, the potential use of GUG-β-CDE (G2) having glucose as a spacer between PAMAM dendrimer and CD as a novel gene transfer carrier was evaluated [45]. GUG-β-CDE (G2) was found to have lower hemolytic activity than PAMAM dendrimer (G2), suggesting that GUG-β-CDE (G2, DSC1.8) had lower local irritation than PAMAM dendrimer (G2). Of GUG-β-CDEs (G2, DSC1.8) having the various DSC, GUG-β-CDE (G2, DSC1.8) strikingly possessed much higher gene transfer activity than α-CDE (G2, DSC1.2) and β-CDE (G2, DSC1.3) in A549 and RAW264.7 cells. These results suggest a crucial role of a spacer between PAMAM dendrimer and CD for high gene transfer activity of GUG-β-CDE (G2, DSC1.8). In sharp contrast to linear PEI (10 kDa), GUG-β-CDE (G2, DSC1.8) had negligible cytotoxicity. Hence, these are very important results regarding the development of carriers for pDNA and oligonucleotide drugs, because GUG-β-CDE (G2, DSC1.8) can be simple, efficiently synthesize the conjugates, and sufficiently deliver the drugs while showing only slight cytotoxicity [45]. Additionally, Anno et al. evaluated the GUG-β-CDE (G2, DSC1.8)) as an in vivo gene transfer carrier [46]. Following intravenous injection of the polyplex with pDNA in mice, GUG-β-CDE (G2, DSC1.8) provided higher gene transfer activity than α-CDE (G2, DSC1.2) and β-CDE (G2, DSC1.3) in kidney with negligible changes in blood chemistry values. In conclusion, the present findings suggest that GUG-β-CDE (G2, DSC1.8) has the potential for a novel polymeric pDNA carrier in vitro and in vivo [43]. Collectively, these results suggest that GUG-β-CDE (G2, DSC1.8) could have the potential for a novel gene transfer carrier, compared to α-CDE (G2, DSC1.2), β-CDE (G2, DSC1.3), and PEI [45,46].

In order to clarify the enhancing mechanism for high gene transfer activity of GUG-β-CDE (G2, DCS1.8), Anno et al. investigated the physicochemical properties, cellular uptake, endosomal escape, and nuclear translocation of the pDNA complexes as well as pDNA release from the complexes [47]. The particle size, ζ-potential and cellular uptake of GUG-β-CDE (G2, DSC1.8)/pDNA complex were mostly comparable to those of α-CDE (G2, DS1.2) and β-CDE (G2, DSC1.3). Meanwhile, GUG-β-CDE (G2, DSC1.8)/pDNA complex was likely to have a high endosomal escaping ability and nuclear localization ability in A549 and RAW264.7 cells. It should be noted that the pDNA condensation and decondensation abilities of GUG-β-CDE (G2, DSC1.8) were lower and higher than that of α-CDE (G2, DSC 1.2) or β-CDE (G2, DSC1.3), respectively. Hence, superior gene transfer activity of GUG-β-CDE (G2, DSC1.8) could be ascribed to a favorable conformation change of pDNA for pDNA release from the conjugate in cells to α-CDE (G2, DSC 1.2) or β-CDE (G2, DSC1.3). These results suggest that high gene transfer activity of GUG-β-CDE (G2, DS 1.8) could be, at least in part, attributed to high endolysosomal escaping ability, nuclear localization ability, and suitable pDNA release from its complex [47].

### 4.2. GUG-β-CDE for siRNA Delivery

Next, to investigate the potentials of GUG-β-CDE (G2) as a siRNA carrier, Anno et al. evaluated the RNAi effect of its complex with siRNA against transthyretin (TTR) mRNA (siTTR) for the treatment of familial amyloidotic polyneuropathy (FAP) [48]. Among the various GUG-β-CDEs (G2) having DSC1.8, 2.5, 3.0, and 5.0, GUG-β-CDE (G2, DSC1.8) showed the highest siTTR transfer activity. GUG-β-CDE (G2, DSC1.8)/siTTR complex showed no cytotoxicity in HepG2 cells. After intravenous administration of GUG-β-CDE (G2, DSC1.8)/siTTR complex to BALB/c mice, TTR mRNA expression was tended to reduce with negligible changes in blood chemistry data. Particle size, ζ-potential, and cellular association of the GUG-β-CDE (G2, DSC1.8) complex with siTTR were almost the same as those of the other CDEs complexes. Meanwhile, GUG-β-CDE (G2, DSC1.8)/siTTR complex showed a high endosomal escaping ability of siTTR in the cytoplasm. These findings suggest the potential of GUG-β-CDE (G2, DS1.8) as a siRNA carrier for the FAP treatment [48]. Table 2 shows a comparison of the features of α-CDE and GUG-β-CD. Given the results show in Table 2, GUG-β-CDE may be better than α-CDE, although the former is a little harder to obtain.

Next, to enhance interaction between GUG-β-CDE (G2) and siRNA and its RNAi effects of siRNA, Abdelwahab et al. [49] newly synthesized GUG-β-CDE (G3) having the various DSC of 1.6, 3.0, 3.7, 5.0, and 8.6, because the number of positively charged NH_2_ group of GUG-β-CD (G3) is two-fold higher than that of GUG-β-CD (G2). GUG-β-CDEs (G3) formed the positively charged and nano-order complexes with siRNA. Of the siRNA complexes with five GUG-β-CDEs (G3), the complex with GUG-β-CDE (G3, DSC3.7) showed the highest RNAi effect and cellular uptake with negligible cytotoxicity in KB cells at a charge (N/P) ratio of 20. In addition, the RNAi effect and cellular uptake of the complex with GUG-β-CDE (G3, DSC3.7) were higher than those of α-CDE (G3, DSC2.4) and comparable to those of Lipofectamine^®^2000. Furthermore, the complex with GUG-β-CDE (G3, DSC3.7) possessed the endolysosomal escaping ability, the releasing property of siRNA in the cytoplasm, and serum resistance. These results suggest that GUG-β-CDE (G3, DSC3.7) has the potential as a novel siRNA carrier [49].

## 5. Glactose- and Lactose-Appended α-CDE for pDNA and siRNA Delivery

### 5.1. Glactose- and Lactose-Appended α-CDE for Gene Delivery

Achieving targeted pDNA and oligonucleotides delivery that is restricted to relevant tissues and cells in vivo is expected to reduce the dose requirements as well as minimize their toxicities. Therefore, Arima et al. designed and evaluated various CDEs. Figure 6 shows the representative multifunctional CDEs, especially α-CDEs for carriers for pDNA and oligonucleotides.

It is well known that ASGP-R on hepatocytes recognized a galactose moiety; therefore, galactose, lactose, GalNAc, and asialofetuin have been used for targeting ligand to hepatocytes [50]. As mentioned above, givosiran as API of Givlaari^®^ is a GalNAc-conjugate for targeting siRNA to hepatocytes [6]. First of all, to improve in vitro gene transfer efficiency and/or achieve cell-specific gene delivery of α-CDE (G2, DSC1), Wada et al. prepared α-CDE (G2, DSC1) bearing galactose (Gal-α-CDE (G2, DSC1, DSG4)) with the various DS values of the galactose moiety (DSG) as a novel nonviral vector [51]. Gal-α-CDE (G2, DSC1, DSG4) was prepared by attached to primary amino residues of α-CDE (G2, DSC1) using α-D-galactopyranosylphenyl isothiocyanate. The agarose gel electrophoretic studies revealed that Gal-α-CDE (G2, DSC1, DSG4) formed complexes with pDNA and protected the degradation of pDNA by DNase I, but these effects impaired as the DSG value increased. PAMAM dendrimer (G2) and Gal-α-CDE (G2, DSC1, DSG4) exerted pDNA condensation through the complexation, but Gal-α-CDE (G2, DSC1, DSG4) did not. Unexpectedly, Gal-α-CDE (G2, DSC1, DSG4) was found to have much higher gene transfer activity than PAMAM dendrimer (G2) and α-CDE (G2, DSC1) in HepG2, NIH3T3, and A549 cells, which are independent of the expression of ASGP-R. Additionally, transfection activity of Gal-α-CDE (G2, DSC1, DSG4) was insensitive to the existence of competitors (asialofetuin and galactose) and serum. Thus, it is evident that the mechanism for the enhancing effect of Gal-α-CDE (G2, DSC1, DSG4) on gene transfer activity may be due to other factors such as changes in intracellular trafficking and/or stability of pDNA but not the cell surface galactose-specific receptor. In addition, the only very slight recognition ability of Gal-α-CDE (G2, DSC1, DSG 4) to ASGP-R may be attributed to low DSG values and short length between primary amino groups of PAMAM dendrimers (G2) and the galactose moiety. Actually, as described below, in the case of Man-α-CDEs (G2, G3) using the similar ligand and spacer (α-D-mannosylphenyl isothiocyanate) to α-D-galactopyranosylphenyl isothiocyanate of Gal-α-CDE (G2, DSC1, DSG 4), they were not able to recognize mannose receptor (MR); therefore, the compounds would not be adequate for spacers of ASGP-R and MR. Taken together, attention should be paid to the type of spacer and the number of ligands in design for targeting carriers. These results suggest the potential use of Gal-α-CDE (G2, DSC1, DSG4) as a pDNA carrier in various cells [51].

Next, to improve the targeting ability of α-CDE (G2, DSC1) to hepatocytes further, Arima et al. prepared α-CDE (G2) bearing lactose (Lac-α-CDE (G2, DSC1)) with various DS values of the lactose moiety (DSL) as a novel hepatocyte-selective carrier in hepatocytes. Herein, the reason why lactose was used for targeting ligand is that a disaccharide composed of galactose and glucose, which has a different spacer from α-D-galactopyranosylphenyl isothiocyanate. Fortunately, Lac-α-CDE (G2, DSC1, DSL2.6) was found to have much higher gene transfer activity than PAMAM dendrimer (G2), α-CDE (G2, DSC1), Lac-α-CDE (G2, DSC1, DSL1.2, 4.6, 6.2 and 10.2), and lactosylated dendrimer (Lac-dendrimer, DSL2.4) in HepG2 cells, which are dependent on the expression of cell-surface ASGP-R, reflecting the cellular association of the pDNA complexes [52]. The physicochemical properties of the pDNA complex with Lac-α-CDE (G2, DSC1, DSL2.6) were almost comparable to that with α-CDE (G2, DSC1). Lac-α-CDE (G2, DSC1, DSL2.6) provided negligible cytotoxicity up to a charge (N/P) ratio of 150 in HepG2 cells. Lac-α-CDE (G2, DSL2.6) provided gene transfer activity higher than jetPEI^®^-Hepatocyte to hepatocytes with much less changes in blood chemistry values 12 h after intravenous administration in mice. These results suggest the potential use of Lac-α-CDE (G2, DSC1, DSL2.6) as a nonviral vector for gene delivery toward hepatocytes [52].

In order to achieve much better in vitro gene delivery, the efficiency of α-CDE (G3, DSC2.4) bearing lactose (Lac-α-CDE (G3, DSC1)) with various DSL values as a novel hepatocyte-selective carrier was studied. Lac-α-CDE (G3, DSC2.4, DSL1.2) was found to have much higher gene transfer activity than α-CDE (G3, DSC2.4), Lac-α-CDE (G2, DSC1, DSL2.6), and Lac-α-CDEs (G3, DSC2.4, DSL2.6, 4.1 and 6.1) in HepG2 cells, which are dependent on the expression of ASGP-R on hepatocytes. Lac-α-CDE (G3, DSC2.4, DSL1.2) provided negligible cytotoxicity up to a charge (N/P) ratio of 100 (carrier/pDNA) in HepG2 cells. These results suggest the potential use of Lac-α-CDE (G3, DSC2.4, DSL1.2) as a nonviral vector for gene delivery toward hepatocytes [53].

### 5.2. Galactose- and Lactose-Appended α-CDE for siRNA Delivery

The liver is the largest organ in the body, and hepatic diseases cause more than 1 million deaths each year around the world. As described above, considering the siRNA therapeutics for FAP caused by the deposition of variant transthyretin (TTR) in various organs, hepatocyte-selective siRNA delivery is desired because TTR is predominantly synthesized by hepatocytes. Actually, Onpattoro^®^ (sodium patisiran) is a siRNA drug using DDS technology for the treatment of amyloidogenic transthyretin (ATTR)-FAP [5]. Regarding the CDE study as the development of hepatocyte-specific siRNA carrier, Hayashi et al. reported the potential use of Lac-α-CDE (G3, DSC2.4, DSL1.2) as novel hepatocyte-selective siRNA carriers, and the RNAi effect of siRNA complex with Lac-α-CDE (G3, DSC2.4, DSL1.2) both in vitro and in vivo was evaluated [54,55]. Lac-α-CDE (G3, DSC2.4, DSL1.2)/siRNA complex had the potent RNAi effect against TTR gene expression through adequate physicochemical properties, ASGP-R-mediated cellular uptake, efficient endolysosomal escaping, and the delivery of the siRNA complex to the cytoplasm, but not the nucleus, with negligible cytotoxicity. Lac-α-CDE (G3, DS2.4, DSL1.2)/siRNA complex had the potential to induce the in vivo RNAi effect after intravenous administration in the liver of mice. The blood chemistry values in the α-CDE (G3, DSC2.4) and Lac-α-CDE (G3, DSC2.4, DSL1.2) systems were almost equivalent to those in the control system (5% mannitol solution). Taken together, these results suggest that Lac-α-CDE (G3, DSC2.4, DSL1.2) has the potential for a novel hepatocyte-selective siRNA carrier in vitro and in vivo and has a possibility as a therapeutic tool for FAP to the liver transplantation [54,55].

### 5.3. PEG-Lac-α-CDE (G3) for pDNA and siRNA Delivery

It is well acknowledged that modifications of proteins, oligonucleotides, and nanoparticles with PEG are often used for efficient delivery of drugs administered systemically, since PEGylation can protect them from interaction with blood components and sequestration by the reticuloendothelial system (RES) and consequently prolong retention in the blood circulation [35]. However, PEGylation increased the hydrophilicity of their surface, therefore severely decreasing cellular uptake and endosomal escape processes, which is known as the PEG dilemma. Hence, the proper design of PEG is a must for the development of DDS carriers. To develop a novel hepatocyte-selective gene carrier having long circulation property in blood, Hayashi et al. prepared PEG-Lac-α-CDEs (G3, DSC2.0, DSL1.2, DSP2.1) and evaluated gene delivery efficiency of these conjugates in vitro and in vivo [53]. PEG-Lac-α-CDE (G3, DSC2.0, DSL1.2, DSP2.1) showed higher gene transfer activity than other PEG-Lac-α-CDEs (G3, DSC2.0, DSL1.2, DSP4.0 and 6.2) in HepG2 cells, expressing ASGP-R, and the activity decreased in HeLa cells, nonexpressing the receptor and in the presence of asialofetuin. High gene transfer activity of PEG-Lac-α-CDE (G3, DSC2.0, DSL1.2, DSP2.1) was retained even in the presence of 50% serum, although the activity of Lac-α-CDE (G3, DSC2.0, DSL1.2), which is lacking a PEG moiety, was severely decreased in the presence of 20% serum. PEG-Lac-α-CDE (G3, DSC2.0, DSL1.2, DSP2.1) provided negligible cytotoxicity up to a charge (N/P) ratio of 50 (carrier/pDNA) in HepG2 cells and less acute organ toxicity. PEG-Lac-α-CDE (G3, DSC2.0, DSL1.2, DSP2.1) showed selective gene transfer activity to hepatic parenchymal cells rather than hepatic nonparenchymal cells. These results suggest that PEG-Lac-α-CDE (G3, DSC2.0, DSL1.2, DSP2.1) is useful as a hepatocyte-selective gene carrier in vitro and in vivo [56].

Targeted DDS is required for RNAi therapy to increase the therapeutic effect and to reduce the adverse effect. Especially in TTR-related amyloidosis, hepatocyte-specific delivery is desired because TTR mainly expresses in hepatocytes. Herein, Hayashi et al. reported on a hepatocyte-specific siRNA delivery system using PEG (molecular weight = 2170)-modified lactosylated dendrimer (G3) conjugates with α-CD (PEG-Lac-α-CDE (G3, DSC2, DSL1, DSP2)) for TTR-related FAP therapy and investigated the in vitro and in vivo gene silencing effects of (PEG-Lac-α-CDE (G3, DSC2, DSL1, DEP2)/siRNA polyplexes [57]. PEG-Lac-α-CDE (G3, DSC2, DSL1, DEP2)/TTR siRNA (siTTR) polyplex exhibited ASGPR-mediated cellular uptake, high endolysosomal escaping ability, and localization of the siRNA in the cytoplasm, resulting in significant TTR silencing in HepG2 cells. In addition, Figure 7 shows the interaction of PEG-Lac-α-CDE (G3, DSC2, DSL1, DEP2) and its siRNA polyplex with Peanut agglutinin, a galactose binding lectin. The dissociation constant of the PEG-Lac-α-CDE (G3, DSC2, DSL1, DEP2)/siRNA complex was significantly higher than that of PEG-Lac-α-CDE (G3, DSC2, DSL1, DEP2) alone. Hence, regarding a binding to ASGP-R, sugar cluster effects of the complex may be exerted, leading to its efficient targeting ability of the complex. In vivo studies showed that PEG-Lac-α-CDE (G3, DSC2, DSL1, DEP2)/siTTR polyplex led to a significant TTR-silencing effect in the liver after systemic administration to mice. Moreover, safety evaluation revealed that PEG-Lac-α-CDE (G3, DSC2, DSL1, DEP2)/siTTR polyplex had no significant toxicity, both in vitro and in vivo. These findings suggest the utility of PEG-Lac-α-CDE (G3, DSC2, DSL1, DEP2) as a promising hepatocyte-specific siRNA delivery system, both in vitro and in vivo, and as a therapeutic approach for TTR-related amyloidosis [57].

## 6. Mannose-Appended α-CDE for pDNA and siRNA Carrier

### 6.1. Mannose-Appended α-CDE for pDNA Carrier

MR is a highly effective endocytic receptor with a broad binding specificity encompassing ligands of microbial and endogenous origin and a poorly characterized ability to modulate cellular activation [58]. The human MR expressed on macrophages and hepatic endothelial cells scavenges released lysosomal enzymes, glycopeptide fragments of collagen, and pathogenic micro-organisms, thus reducing damage following tissue injury [59]. The receptor binds mannose, fucose, or N-acetylglucosamine (GlcNAc) residues on these targets [60]. In addition, human tumor-associated macrophages express various CLRs and, most prominently, the MR/CD206, which can be utilized for drug targeting [61].

To improve the activity and the cell specificity of gene transfer of PAMAM dendrimer (G2), Arima et al. prepared α-CDE (G2, DSC1) bearing mannose (Man-α-CDE) having various DS of mannose residue (DSM) using α-D-mannosylphenyl isothiocyanate. Man-α-CDEs (G2, DSC1, DSM1, 3 and 5) formed polyplexes with pDNA, but Man-α-CDE (G2, DSC1, DSM8) did not [62]. Gene transfer activity of Man-α-CDEs (G2, DSC1, DSM1, 3, and 5) and α-CDE (G2, DSC1) were augmented with an increase in the charge (N/P) ratio of carrier/pDNA, without showing cytotoxicity. Contrary to expectations, or as expected from the results of Gal-α-CDEs (G2), Man-α-CDE (G2, DSC1, DSM3, and 5) showed higher gene transfer activity than α-CDE (G2, DSC1) in A549 cells, which recognize mannose, but Man-α-CDE (G2, DSC1, DSM1 and 8) showed almost comparable gene transfer activity to α-CDE (G2, DSC1). On the other hand, no appreciable enhancing effect of Man-α-CDE (G2, DSC1, DSM3, and 5) on the transfer activity was observed in NIH3T3 cells, which do not recognize mannose. However, in the study, cells overexpressing MR such as macrophages were not used. Therefore, further MR-mediated delivery of Man-α-CDEs is needed [62].

To next evaluate in vitro and in vivo gene delivery efficiency of Man-α-CDE (G2, DSC1.1) with the various DSM values as a novel nonviral vector in a variety of cells. Man-α-CDEs (G2, DSC1.1, DSM3.3 and 4.9) were found to have much higher gene transfer activity than PAMAM dendrimer (G2), α-CDE (G2, DSC1.1), and Man-α-CDE (G2, DSC1.1, DSM1.1 and 8.3) in various cells, which are independent of the expression of cell surface MRs [60]. Cellular association of pDNA complexes with PAMAM dendrimer (G2), α-CDE (G2, DSC1.1), and Man-α-CDE (G2, DSC1.1, DSM3.3), and their cytotoxic effects differed only very slightly. Surface plasmon resonance study demonstrated that the specific binding activity of Man-α-CDE (G2, DSC1.1, DSM3.3) to concanavalin A, a plant lectin that binds to the mannose residues of various glycoproteins, was not very strong. Much more conjugation of the mannose moiety to α-CDE (G2, DSC1.1) provided unfavorable physicochemical properties of pDNA complexes for gene transfer, e.g., the low interaction with pDNA, the low enzymatic stability of pDNA, and the lack of pDNA compaction. Man-α-CDE (G2, DSC1.1, DSM3.3) provided gene transfer activity higher than PAMAM dendrimer (G2) and α-CDE (G2, DSC1.1) in kidney 12 h following intravenous injection in mice. These results suggest the potential use of Man-α-CDE (G2, DSC1.1, DSM3.3) as a nonviral vector in an MR-independent manner [63].

To enhance gene transfer activity of Man-α-CDE (G2, DSC1.1, DSM3.3), Arima et al. prepared Man-α-CDE (G3, DSC2.2) with DSM of 5, 10, 13 and 20, and compared their cytotoxicity and gene transfer activity and elucidated the enhancing mechanism for the activity [64,65]. Of the various carriers used here, Man-α-CDE (G3, DSC2, DSM10) provided the highest gene transfer activity in NR8383, a rat alveolar macrophage cell line, A549, NIH3T3, and HepG2 cells, being independent of the expression of MRs. Gene transfer activity of Man-α-CDE (G3, DSC2, DSM10) was not decreased by the addition of 10% serum in A549 cells. Cytotoxicity of the polyplex with Man-α-CDE (G3, DSC2, DSM10) was not observed in A549 and NIH3T3 cells up to a charge (N/P) ratio of 200/1 (carrier/pDNA). The gel mobility and particle size of polyplex with Man-α-CDE (G3, DSC2, DSM10) were relevant to those with α-CDE (G3, DSC2), but ζ-potential, DNase I stability, pDNA condensation of the former polyplex were somewhat different from those of the latter one. Cellular association of polyplex with Man-α-CDE (G3, DSC2, DSM10) was almost comparable to those with PAMAM dendrimer (G3) and α-CDE (G3, DSC2). The addition of mannan and mannose attenuated gene transfer activity of Man-α-CDE (G3, DSC2, DSM10) in A549 cells. Alexa-pDNA complex with TRITC-Man-α-CDE (G3, DSC2, DSM10), but not the complex with TRITC-α-CDE (G3, DSC2), was found to translocate to the nucleus at 24 h after incubation in A549 cells. Interestingly, the HVJ-E vector including mannan, but neither the vector alone nor the vector including dextran, suppressed the nuclear localization of TRITC-Man-α-CDE (G3, DSC2, DSM10) to a striking degree after 24 h incubation in A549 cells. These results suggest that Man-α-CDE (G3, DSC2, DSM10) has less cytotoxicity and prominent gene transfer activity through its serum-resistant and endosome-escaping abilities as well as nuclear localization ability, although it does not possess MR-mediated gene targeting ability [64,65].

### 6.2. Mannose-Appended α-CDE for siRNA Delivery

Man-α-CDE (G2, G3) did not show enough MR-mediated internalization to APC, probably due to (1) a short distance of the spacer which linked mannose to the dendrimer of Man-α-CDEs (G2, G3) and (2) poor flexibility of spacer composed of phenylisothiocyanate. Recently, Kovacs et al. reported that MR 1 expression does not determine the uptake of high-density mannose-PAMAM dendrimers by activated macrophage populations. In this study, mannose-PAMAM dendrimer was synthesized using PAMAM dendrimer (G5) and α-D-mannopyranosylphenyl isothiocyanate, suggesting the spacer seems to be inadequate for targeting MR [66]. Therefore, to solve those problems, Arima et al. newly designed thioalkylated mannose-modified α-CDE (Man-S-α-CDE (G3)), having longer spacer and higher flexibility, compared with the Man-α-CDE system, and evaluated the in vitro RNAi effect of the siRNA complex with Man-S-α-CDE (G3). Herein, a new spacer, that is, 1-α-d-mannosyl-oxypropyl-thio-ethyl-carboxylic acid (thioalkylated mannose) was used. Motoyama et al. designed and evaluated the potential use of thioalkylated mannose-modified PAMAM dendrimer (G3) conjugates with α-CD (Man-S-α-CDE (G3, DSC2)) as novel antigen-presenting cell (APC)-selective siRNA carriers [67]. In the study, siGL2 and siGL3, siRNAs for reporter genes, were used. Man-S-α-CDE (G3, DSC2, DSM4)/siRNA complex had the potent RNAi effects in both NR8383 cells, a rat alveolar macrophage cell line, and JAWSII cells, a mouse dendritic cell line, through adequate physicochemical properties, MR-mediated cellular uptake, and efficient phagosomal escape of the siRNA complex [67]. In the study, a spacer was changed from α-D-mannosylphenyl isothiocyanate to 1-α-D-mannosyl-oxypropyl-thio-ethyl-carboxylic acid, consequently resulting in MR-mediated siRNA delivery by Man-S-α-CDE (G3, DSC2, DSM4). Hence, importance of a spacer type for MR-mediated siRNA delivery of Man-S-α-CDEs (G3, DSC2, DSM4) was evidently shown in the study. In addition, cytotoxic activities of the siRNA complexes with α-CDE (G3, DSC2) and Man-S-α-CDE (G3, DSC2, DSM4) were almost negligible up to a charge (N/P) ratio of 100 (carrier/siRNA). Taken together, these results suggest that Man-S-α-CDE (G3, DSC2, DSM4) has the potential for a novel APC-selective siRNA carrier [64].

Fulminant hepatitis is a serious, life-threatening disorder and is associated with inflammatory cytokines produced by Kupffer cells [68]. However, a number of clinical trials for the treatment of fulminant hepatitis did not show enough substantial benefits. Since NF-*κ*B is a key mediator of inflammatory response in Kupffer cells, NF-*κ*B and its associated pathways are complicatedly concerned about hepatic homeostasis [69]. Discriminating inhibition of NF-*κ*B signaling has been expected to treat various liver diseases including fulminant hepatitis. To clarify the potential use of Man-S-α-CDE (G3, DSC2, DSM4) as a novel APC-specific siRNA carrier, we evaluated the RNAi effect of NF-*κ*B p65 siRNA (sip65) complex with Man-S-α-CDE (G3, DSC2, DSM4) both in vitro and in vivo [70]. Man-S-α-CDE (G3, DSC2, DSM4)/sip65 complex significantly suppressed NF-*κ*B p65 mRNA expression and nitric oxide production from lipopolysaccharide (LPS)-stimulated NR8383 cells by adequate physicochemical properties and MR-mediated cellular uptake. Intravenous injection of Man-S-α-CDE (G3, DSC2, DSM4)/sip65 complex extended the survival rate of LPS-induced fulminant hepatitis model mice. In addition, intravenous administration of Man-S-α-CDE (G3, DSC2, DSM4)/sip65 complex had the potential to induce the in vivo RNAi effect by significant suppression of mRNA expression of NF-*κ*B p65 and inflammatory cytokines in the liver of fulminant hepatitis model mice induced by LPS/D-galactosamine (D-Gal) without any significant side effects. Additionally, the serum levels of enzymes were significantly attenuated by injection of Man-S-α-CDE (G3, DSC2, DSM4)/sip65 complex in fulminant hepatitis model mice. However, to exert the RNAi effect, preadministration of Man-S-α-CDE (G3, DSC2, DSM4)/sip65 complex is necessary because of late onset of inflammatory cytokine expression in an initial state; therefore, a modality without the need for pretreatment is required. Collectively, these results suggest that Man-S-α-CDE (G3, DSC2, DSM4) has the potential as a novel APC-selective sip65 carrier for the treatment of LPS/D-Gal-induced fulminant hepatitis in mice [70].

## 7. Fucose-Appended α-CDE for Decoy DNA Delivery

Since NF-*κ*B is a key mediator of inflammatory response in Kupffer cells, NF-*κ*B decoy would be an attractive candidate for the treatment of fulminant hepatitis. Recently, Opanasopit et al. revealed that fucosylated protein is preferentially taken up by Kupffer cells via a fucose receptor (Fuc-R) [71]. Additionally, as described above, MR recognizes both mannose and fucose [71]. Therefore, the fucosylation to NF-*κ*B decoy carrier is one of the prominent approaches for Kupffer cell-selective delivery. Akao et al. reported on treating LPS-induced fulminant hepatitis by NF-*κ*B decoy complex with fucose-appended PAMAM dendrimer (G2) conjugate with α-CD (Fuc-S-α-CDE (G2, DSC1, degree of substitution of fucose (DSF)2)) [72,73]. Fuc-S-α-CDE (G2, DSC1, average DS of fucose (DSF2))/NF-*κ*B decoy complex significantly suppressed nitric oxide and tumor necrosis factor-α (TNF-α) production from LPS-stimulated NR8383 cells, a rat alveolar macrophage cell line, by adequate physicochemical properties and Fuc-R-mediated cellular uptake. Intravenous injection of Fuc-S-α-CDE (G2, DSC1, DSF2)/NF-*κ*B decoy complex extended the survival of LPS-induced fulminant hepatitis model mice. In addition, Fuc-S-α-CDE (G2, DSC1, DSF2)/NF-*κ*B decoy complex administered intravenously highly accumulated in the liver, compared to naked NF-*κ*B decoy alone. Additionally, the liver accumulation of Fuc-S-α-CDE (G2, DSC1, DSF2)/NF-*κ*B decoy complex was inhibited by the pretreatment with gadolinium(III) chloride (GdCl_3_), a specific inhibitor of Kupffer cell uptake. Moreover, the serum aspartate aminotransferase (AST), alanine aminotransferase (ALT), and TNF-α levels in LPS-induced fulminant hepatitis model mice were significantly attenuated by the treatment with Fuc-S-α-CDE (G2, DSC1, DSF2)/NF-*κ*B decoy complex, compared with naked NF-*κ*B decoy alone. Furthermore, Fuc-S-α-CDE (G2, DSC1, DSF2)/NF-*κ*B decoy complex would be superior to Fuc-S-α-CDE (G2, DSC1, DSF2)/NF-*κ*B siRNA complex in terms of the presence or absence of pretreatment. Taken together, these results suggest that Fuc-S-α-CDE (G2, DSC1, DSF2) has the potential for a novel Kupffer cell-selective NF-*κ*B decoy carrier for the treatment of LPS-induced fulminant hepatitis in mice [72,73].

## 8. Folate-Appended α-CDE

### 8.1. Folate-Appended α-CDE for pDNA Delivery

Almost 1.9 million new cancer cases are expected to be diagnosed in 2021, and approximately 608,570 Americans are expected to die of cancer in 2021, which translates to about 1670 deaths per day [74]. Hence, cancer is the second most common cause of death in the US, exceeded only by heart disease [74]. Meanwhile, folate receptors (FRs) are glycoproteins with molecular weights of 38–44 kDa, which exist in several isoforms [75]. Of four human isoforms of FR (α, β, γ, and δ), FR-α is expressed at a few sites of normal epithelial membranes for instance in the proximal tubule cells of the kidneys and in a variety of solid cancer types of epithelial origin [76]. FRs have a high affinity for folate in the subnanomolar range (K_d_ ≈ 1–10 nM), compared with folate transporters, which have a low affinity for folate in the micromolar range [77].

As described above, a folate is important for producing and maintaining new cells because it can participate in nucleotide synthesis. In addition, only the malignant cells, not normal cells, transport folate-conjugates; thus, the folate–drug and folate–carrier conjugation can improve tumor-targeted drug delivery. Meanwhile, PEG prevents the opsonin binding to the nanoparticle surfaces and, consequently, recognition as well as phagocytosis of the nanoparticles by the mononuclear phagocytic system, which enhances the blood circulation time. Based on the background, Arima et al. provided remarkable aspects as novel carriers for pDNA and siRNA [78]. To develop novel α-CDEs with tumor cell specificity, Arima et al. prepared folate-appended α-CDEs (Fol-α-CDEs) and folate-PEG-appended α-CDEs (Fol-PEG-α-CDE (G3, DSC2.4) with the various DS of folate (DSF) and evaluated in vitro and in vivo gene transfer activity, cytotoxicity, cellular association, and physicochemical properties. In vitro gene transfer activity of Fol-α-CDEs (G3, DSC2.4, DSF2, 5, and 7) was lower than that of α-CDE (G3, DSC2.4) in KB cells, FR-overexpressing cancer cells. Of the three Fol-α-CDEs (G3, DSC2.4, DSF2, 5 and 7), Fol-PEG-α-CDE (G3, DSC2.4, DSF5, DSP5) had the highest gene transfer activity in KB cells. The activity of Fol-PEG-α-CDE (G3, DSC2.4, DSF5, DSP5) was significantly higher than that of α-CDE (G3, DSC2.4) in KB cells, but not in A549 cells, FR-negative cells. Negligible cytotoxicity of the pDNA complex with Fol-PEG-α-CDE (G3, DSC2.4, DSF5, DSP5) was observed in KB cells or A549 cells up to a charge (N/P) ratio of 100/1 (carrier/pDNA). The cellular association of the pDNA complex with Fol-PEG-α-CDE (G3, DSC2.4, DSF5, DSP5) could be mediated by FR-α on KB cells, resulting in its efficient cellular uptake. Fol-PEG-α-CDE (G3, DSC2.4, DSF5, DSP5) had a higher binding affinity with folate-binding protein than α-CDE (G3, DSC2.4), although the physicochemical properties of pDNA complex with Fol-PEG-α-CDE (G3, DSC2.4, DSF5, DSP5) were almost comparable to that with α-CDE (G3, DSC2.4), although the onset charge (N/P) ratio and the compaction ability of Fol-PEG-α-CDE (G3, DSC2.4, DSF5, DSP5) were slightly different. Fol-PEG-α-CDE (G3, DSC2.4, DSF5, DSP5) tended to show higher gene transfer activity than α-CDE (G3, DSC2.4) 12 h after intratumoral injection in mice. These results suggest that Fol-PEG-α-CDE (G3, DSC2.4, DSF5, DSP5), not Fol-α-CDEs (G3, DSC2.4, DSF5), could be potentially used as an FR-overexpressing cancer cell-selective pDNA carrier [78].

### 8.2. Fol-PEG-α-CDE for siRNA Delivery

Systemic siRNA therapy warrants the development of clinically suitable, safe, and effective drug delivery systems [79]. Arima et al. next investigated the potential use of Fol-PEG-α-CDE (G3, DSC2, DSF2, 4 and 7, DSP2, 4, and 7) with various DSF as a tumor-selective siRNA carrier to FR-overexpressing cancer cells in vitro and in vivo [80]. Of the three Fol-PEG-α-CDEs (G3, DSC2.4, DSF2, 4 and 7, DSP2, 4, and 7), Fol-PEG-α-CDE (G3, DSC2.4, DSF4, DSP4) had the highest siRNA transfer activity in KB cells (FR-α positive). Fol-PEG-α-CDE (G3, DSC2.4, DSF4, DSP4) was endocytosed into KB cells through FR-α. No cytotoxicity of the siRNA complex with Fol-PEG-α-CDE (G3, DSC2.4, DSF4, DSP4) was observed in KB cells (FR-α positive) or A549 cells (FR-α negative) up to a charge (N/P) ratio of 100/1 (carrier/siRNA). In addition, the siRNA complex with Fol-PEG-α-CDE (G3, DSC2.4, DSF4, DSF4) showed neither interferon response nor inflammatory response. Importantly, the siRNA complex with Fol-PEG-α-CDE (G3, DS2.4, DSF4, DSP4) tended to show the in vivo RNAi effects after intratumoral injection and intravenous injection in tumor cells-bearing mice. The FITC-labeled siRNA and TRITC-labeled Fol-PEG-α-CDE (G3, DSC2.4, DSF4, DSP4) were actually accumulated in tumor tissues after intravenous injection in the mice. In conclusion, the present results suggest that Fol-PEG-α-CDE (G3, DSC2.4, DSF4, DSP4) could potentially be used as an FR-overexpressing cancer cell-selective siRNA delivery carrier in vitro and in vivo [80]. However, Fol-PEG-α-CDE (G3, DSC2.4, DSF4, DSP4)/siRNA complex did not induce a significant in vivo RNAi effect after intravenous administration to tumor-bearing mice, possibly resulting from immediate dissociation of the complex in blood. Herein, to develop the novel siRNA carrier having high blood circulating ability, high in vivo siRNA transfer activity, and high safety profile, Ohyama et al. newly prepared Fol-PEG-α-CDEs (G4, DSC2.9, DSF2, DSP2) and evaluated their potential as tumor-targeting siRNA carriers in vitro and in vivo [81]. Fol-PEG-α-CDE (G4, DSC2.9, DSF2, DSP2)/siRNA complex had the prominent RNAi effect through adequate physicochemical properties, FR-α-mediated endocytosis, efficient endolysosomal escape, and siRNA delivery to the cytoplasm with negligible cytotoxicity. Actually, Fol-PEG-α-CDEs (G4, DSC2.9, DSF2, DSP2)/siRNA complex as well as Fol-PEG-α-CDEs (G3, DS2.4, DSF4, DSP4)/siRNA complex showed negligible cytotoxicity, although PAMAM dendrimer (G4)/siRNA complex and α-CDE (G4)/siRNA complex provided severe cytotoxicity to KB cells (FR-positive cells) (Figure 8).

From the viewpoint of biological fates of Fol-PEG-α-CDE (G4, DSC2.9, DSF2, DSP2)/siRNA complex and Fol-PEG-α-CDE (G3, DSC2.4, DSF4, DEP4)/siRNA complex, the time courses of blood levels of FITC-labeled siRNA, TRITC-labeled Fol-PEG-α-CDE (G4, DSC2.9, DSF2, DSP2) and TRITC-labeled Fol-PEG-α-CDE (G3, DSC2.4, DSF4, DEP4) were determined after intravenous injection of the complexes to BALB/c mice bearing Colon-26 tumor cells (Figure 9). The half-life of FITC-siRNA in the Fol-PEG-α-CDE (G4, DSC2.9, DSF2, DSP2)/siRNA complex system was seven times higher than the Fol-PEG-α-CDE (G3, DSC2.4, DSF4, DEP4)/siRNA complex system. Similarly, the half-life of TRITC-labeled Fol-PEG-α-CDE (G4, DSC2.9, DSF2, DSP2) was also seven times higher than that of Fol-PEG-α-CDE (G3, DSC2.4, DSF4, DEP4). These results suggest that Fol-PEG-α-CDE (G4, DSC2.9, DSF2, DSP2)/siRNA complex showed longer retention in blood than Fol-PEG-α-CDE (G3, DSC2.4, DSF4, DEP4)/siRNA complex, probably due to high serum stability.

Importantly, Fol-PEG-α-CDE (G4, DSC2.9, DSF2, DSP2) improved in vivo RNAi effects of siRNA, compared to Fol-PEG-α-CDE (G3, DSC2.4, DSF4, DSP4). In fact, Fol-PEG-α-CDE (G4, DSC2.9, DSF2, DSP2) complex with siRNA against Polo-like kinase 1 (siPLK1) suppressed the tumor growth, compared to the control siRNA complex. The potent RNAi effects of Fol-PEG-α-CDE (G4, DSC2.9, DSF2, DSP2)/siPLK1 complex could be ascribed to the high stability of the complex in blood, resulting in efficient delivery to tumor tissue and tumor cells. However, to exert the RNAi effects, multiple administration of the complex is needed. Therefore, further study is needed to improve delivery efficiency. These results suggest that Fol-PEG-α-CDE (G4, DSC2.9, DSF2, DSP2) has the potential as a novel tumor-targeting siRNA carrier in vitro and in vivo [78]. Figure 10 and Table 3 summarize the comparison of Fol-PEG-α-CDE (G3, DSC2.4, DSF4, DSP4) and Fol-PEG-α-CDE (G4, DSC2.9, DSF2, DSP2) as siRNA carriers. Overall, Fol-PEG-α-CDE (G4, DSC2.9, DSF2, DSP2) should be better than Fol-PEG-α-CDE (G3, DSC2.4, DSF4, DSP4) as a siRNA carrier for cancer treatment, although further detailed study is necessary. Furthermore, Arima and colleague revealed that Fol-PEG-α-CDE (G4, DSC2.9, DSF2, DSP2) has the potential for microRNA carrier to treat cancers (unpublished data).

### 8.3. Fol-PEG-GUG-β-CDE for siRNA Delivery

Thus far, Arima et al. previously reported the utility of GUG-β-CDEs (G3) as siRNA carriers [82]. To further improve the potency of GUG-β-CDEs (G3), Mohamed et al. examined whether Fol-PEG-GUG-β-CDEs (G3, DSC3.7) having DSF of 3.9, 6.7, and 7.3 possess the potential for the utility as tumor-selective siRNA carriers [83]. Of various Fol-PEG-GUG-β-CDEs (G3, DSC3.7, DSF3.9, 6.7, and 7.3, DSP3.9, 6.7, and 7.3), Fol-PEG-GUG-β-CDE (G3, DSC3.7, DSF6.7, DSP6.7) showed the highest siRNA transfection activity at a charge (N/P) ratio of 50 (carrier/siRNA) in both 786-0-luc cells (FR-α (+)), a luciferase stably expressing human renal cancer cell line, and KB cells (FR-α (+)). In addition, the cellular uptake of the complex was significantly decreased by the addition of folic acid in a concentration-dependent manner, suggesting its FR-α-mediated endocytosis pathway. Moreover, Fol-PEG-GUG-β-CDE (G3, DSC3.7, DSF6.7, DSP6.7)/siRNA complex induced a potent RNAi effect, comparable to Lipofectamin^®^2000/siRNA complex. Furthermore, Fol-PEG-GUG-β-CDE (G3, DSC3.7, DSF6.7, DSP6.7) complex with siPLK1 showed significant cytotoxic activity in KB cells. Thus, Fol-PEG-GUG-β-CDE (G3, DSC3.7, DSF6.7, DSP6.7) has the potential as a targeted siRNA delivery carrier for FR-α-overexpressing tumor cells [83].

## 9. Ternary Complex of CDE with Low-Molecular-Weight Sacran

### 9.1. Lac-α-CDE with Low-Molecular-Weight Sacran for siRNA Delivery

As described above, PEGylation is a very useful DDS technology. However, it has a drawback termed the PEG dilemma. One promising approach for overcoming this drawback is electrostatic encapsulation or surface modification of cationic nanoparticles with anionic biodegradable natural polymers such as hyaluronic acid, chondroitin sulfate, or polyglutamic acid because such anionic complexes can prevent agglutination with blood components after systemic injection [84]. On the other hand, the Aphanothece sacrum is a freshwater unicellular cyanobacterium that efficiently fixes CO_2_ during aqua cultivation in rivers [85]. Sacran, a polysaccharide extracted from Aphanothece sacrum, composed of various monosaccharide units, e.g., glucose, galactose, mannose, xylose, rhamnose, and fucose, with compositions of 25.9%, 11.0%, 10.0%, 16.2%, 10.2%, and 6.9%, respectively [86]. Additionally, it has 20–25% uronic acids and ~1% arabinose, galactosamine, and muramic acid [87]. The absolute molecular weight of sacran, Mw, measured by using multiangle static light scattering (MALLS) was >10^7^ g/mol [86]. The sacran molecule possesses a large number of sulfate and carboxyl groups and is known to be an anionic macromolecule and has an extremely high molecular weight (approximately 29 MDa) [86]. Sacran has been consumed as a functional food to ameliorate allergic tendency and gastroenteritis in Japan [86]. Hence, sacran is thought to be a safe biomaterial. Additionally, numerous cosmetics containing sacran have been commercially available in Japan [86]. Therefore, sacran is a potentially safe coating material for cationic carrier/siRNA complexes. However, little is known about the suitability of sacran as a siRNA carrier. On the basis of these backgrounds, Hayashi et al. newly developed the ternary complexes consisting of Lac-α-CDE (G2, DSC2, DSL1), siRNA, and the anionic polysaccharide sacrans and evaluated their utility as siRNA transfer carriers [87]. Three kinds of the low-molecular-weight sacrans, i.e., sacran (100) (Mw 44,889Da), sacran (1000) (Mw 943,692Da) and sacran (10,000) (Mw 1,488,281Da) were used. Lac-α-CDE (G2, DSC2, DSL1)/siRNA/sacran ternary complexes were prepared by adding the low-molecular-weight sacrans to the Lac-α-CDE (G2, DSC2, DSL1)/siRNA binary complex solution. Cellular uptake of the ternary complex with sacran (100) was higher than that of the binary complex or the other ternary complexes with sacran (1000) and sacran (10,000) in HepG2 cells. Here, Hayashi et al. employed low-molecular-weight sacrans with different molecular weights since an intact sacran could be too large for intravenous injection [84]. Additionally, the ternary complex possessed high serum resistance and endolysosomal escaping ability in HepG2 cells. High liver levels of siRNA and Lac-α-CDE (G2, DSC2, DSL1) were observed after the intravenous injection of the ternary complex rather than that of the binary complex. Moreover, intravenous injection of the ternary complex induced the significant RNAi effect in the liver of mice with negligible changes in blood chemistry values. Therefore, a ternary complexation of the Lac-α-CDE (G2, DSC2, DSL1)/siRNA binary complex with low-molecular-weight sacran is useful as a hepatocyte-specific siRNA delivery system [87].

### 9.2. Fol-PEG-α-CDE with Low-Molecular-Weight Sacran for siRNA Delivery

The potential of the sacrans was evaluated by Ohyama et al. using Fol-PEG-α-CDE (G4, DSC3, DSF2, DSP2) [88], that is, Ohyama et al. prepared ternary complexes of Fol-PEG-α-CDE (G4, DSC3, DSF2, DSP2)/siRNA with low-molecular-weight sacrans to achieve more effective siRNA transfer activity (Figure 11) [88]. Among the different molecular-weight sacrans, i.e., sacran (100), (1000), and (10,000), sacran (100) significantly increased the cellular uptake and the RNAi effects of Fol-PEG-α-CDE (G4, DSC3, DSF2, DSP2)/siPLK1 binary complex with negligible cytotoxicity in KB cells (FR-α positive cells). In addition, the ζ-potential and particle size of Fol-PEG-α-CDE (G4, DSC3, DSF2, DSP2)/siPLK1 complex were decreased by the ternary complexation with sacran (100). Importantly, the in vivo RNAi effect of the ternary complex after the intravenous injection to tumor-bearing BALB/c mice was significantly higher than that of the binary complex. In conclusion, Fol-PEG-α-CDE (G4, DSC3, DSF2, DSP2)/siRNA/sacran (100) ternary complex has potential as a novel tumor-selective siRNA delivery system [85]. As described above, since sacran has unique properties, sacran is a promising new biomaterial for gene and oligonucleotide delivery.

## 10. Co-Delivery of siRNA and Low-Molecular-Weight Antitumor Drug

To overcome chemoresistance using RNAi therapy, the simultaneous co-delivery of two types of siRNAs and of siRNA with antitumor drugs may be required in the combination therapy of multidrug resistance cancer [89]. Doxorubicin (DOX) is a common chemotherapy medication used to treat many kinds of cancers, such as breast cancer, bladder cancer, and lymphoma. DOX interacts with DNA, leading to a break in the DNA strand, which blocks DNA transcription and replication, resulting in apoptosis of cancer cells. Mohammed et al. evaluated the potential of Fol-PEG-GUG-β-CDE (G3, DSC4, DSF6.7, DSP6.7) as a carrier for the low-molecular antitumor drug DOX. Further, to fabricate advanced antitumor agents, Mohammed et al. prepared a ternary complex of Fol-PEG-GUG-β-CDE (G3, DSC4, DSF6.7, DSP6.7) /DOX/siPLK1 and evaluated its antitumor activity both in vitro and in vivo [87]. Fol-PEG-GUG-β-CDE (G3, DSC4, DSF6.7, DSP6.7) released DOX in an acidic pH and enhanced the cellular accumulation and cytotoxic activity of DOX in FR-α-overexpressing KB cells. Importantly, the Fol-PEG-GUG-β-CDE (G3, DSC4, DSF6.7, DSP6.7)/DOX/siPLK1 ternary complex exhibited higher cytotoxic activity than a binary complex of Fol-PEG-GUG-β-CDE (G3, DSC4, DSF6.7, DSP6.7) with DOX or siPLK1 in KB cells. In addition, the cytotoxic activity of the ternary complex was reduced by the addition of folic acid, a competitor against FR-α. Furthermore, the ternary complex showed a significant antitumor activity after intravenous administration to the tumor-bearing mice. These results suggest that Fol-PEG-GUG-β-CDE (G3, DSC4, DSF6.7, DSP6.7) has the potential of a tumor-selective co-delivery carrier for DOX and siPLK1 [90].

## 11. Supramolecular with PEG-Appended CDE as Sustained Release System of pDNA

Nonviral gene delivery suffers from a number of limitations, including short transgene expression times and low transfection efficiency. Supramolecular chemistry is an extremely useful and important domain for understanding pharmaceutical sciences because various physiological reactions and drug activities are based on supramolecular chemistry; especially, CD-based supramolecules are promising systems in the DDS field [91]. Motoyama et al. reported whether PPRXs of PEG (Mw 2,000)-grafted α-CD (α-CD)/PAMAM dendrimer conjugate (PEG-α-CDE (G2, DSC1.5, DSP4) with CDs have the potential for the novel sustained release systems for pDNA. PEG-α-CDE (G2, DSC1.5, DSP4)/pDNA complex formed PPRXs with α-CD and γ-CD solutions but not with β-CD solution [92]. In the PEG-α-CDE (G2, DSC1.5, DSP4)/CDs PPRX systems, 20.6 mol of α-CD and 11.8 mol of γ-CD were involved in the PPRXs formation with one PEG chain by α-CD and γ-CD, respectively, consistent with the PEG-dendrimer (G2, DSP4)/CDs systems. PEG-α-CDE (G2, DSC1.5, DSP4)/pDNA/α-CD PPRX (Figure 12a) and PEG-α-CDE (G2, DSC1.5, DSP4)/pDNA/γ-CD PPRX formed hexagonal and tetragonal columnar channels in the crystalline phase, respectively. In addition, the CDs PPRX provided the sustained release of pDNA from PEG-α-CDE (G2, DSC1.5, DSP4) complex with pDNA at least 72 h in vitro. The release of pDNA from CDs PPRX retarded as the volume of dissolution medium decreased (Figure 12b).

Furthermore, the PEG-α-CDE (G2, DSC1.5, DSP4)/γ-CD PPRX system showed sustained transfection efficiency after intramuscular injection to mice at least for 14 days. These results suggest that the PEG-α-CDE (G2, DSC1.5, DSP4)/CD PPRX systems are useful for novel sustained pDNA release systems [76]. In addition, Arima et al. reported the use of PEG-α-CDE (G2, DSC1, DSP3) as sustained-release carriers for siRNA. PEG-α-CDE (G2, DSC1, DSP3)/siRNA complex showed adequate physicochemical properties and cellular association. siRNA released, at least in part, from the complex in Colon-26 cells and showed the RNAi effect with negligible cytotoxicity [93]. PEG-α-CDE (G2, DSC1, DSP3) /siRNA complex formed water-insoluble PPRXs with α-CD and γ-CD. The PPRXs greatly augmented the encapsulation efficacy of siRNA. Importantly, the release of the PEG-α-CDE (G2, DSC1, DSP3) /siRNA complex from the PPRXs was prolonged at least for 24 h in vitro. Moreover, decreasing the volume of the dissolution medium was concomitant with prolonging the release of the siRNA complex from the PPRXs. Collectively, these findings suggest that PEG-α-CDE (G2, DSC1, DSP3)/CD PPRXs are useful for sustained siRNA release systems [93].

## 12. GUG-β-CDE for Genome Editing

Genome editing technologies have great potential as tools to facilitate gene therapy for hereditary diseases, by the destruction or repair of the responsible genes [94]. The CRISPR/Cas9 system consists of a Cas9 nuclease complexed with a single guide RNA (sgRNA), the latter containing a complementary sequence of the target DNA sequence [95]. In recent years, the co-formulated Cas9 mRNA/sgRNA and preassembled Cas9/sgRNA complex (Cas9 ribonucleoprotein; Cas9 RNP) have attracted considerable attention as an alternative method for the introduction of the CRISPR-Cas9 system in the cells [96]. However, Cas9 RNP generally also has low cell membrane permeability; thus, an efficient intracellular delivery system for Cas9 RNP is urgently required [97]. In recent years, several researchers have developed Cas9 RNP delivery systems [98,99]. Therefore, genome editing that occurs within a narrow range of the injection area requires multiple invasive injections in different brain areas. To achieve genome editing across a wide area of the brain, a Cas9 RNP carrier that can be delivered by intraventricular or intrathecal administration and efficiently incorporated into the cells is needed.

As described above, it is of interest that GUG-β-CDE (G3) can interact with genes or siRNAs and with proteins such as transthyretin and albumin; therefore, GUG-β-CDE (G3) is a potential Cas9 RNP carrier. Based on this background, Taharabaru et al. evaluated the potential of GUG-β-CDEs (G3, DSC1.5, 3, and 6.5) as a Cas9 RNP carrier in SH-SY5Y cells, a human neuroblastoma cell line [96]. Moreover, the in vivo genome editing activity in the mouse brain after a single intraventricular administration of the Cas9 RNP ternary complex with GUG-β-CDE (G3, DSC3) was examined to evaluate its potential as a Cas9 RNP carrier, which is able to induce genome editing across a wide area of the brain. As a result, a Cas9 RNP ternary complex with GUG-β-CDE (G3, DSC3) was prepared by only mixing the components. The resulting complex exhibited higher genome editing activity than the complex with PAMAM dendrimer (G3), Lipofectamine^®^3000, or Lipofectamine^®^CRISPRMAX in SH-SY5Y cells. In addition, GUG-β-CDE (G3, DSC3) enhanced the genome editing activity of Cas9 RNP in the whole mouse brain after a single intraventricular administration. Thus, GUG-β-CDE (G3, DSC3) is a useful Cas9 RNP carrier that can induce genome editing in the neuron and brain [99]. Table 4 shows various CDEs introduced in the review.

## 13. CDE as Active Pharmaceutical Ingredients (API)

### 13.1. CDE as an Anti-Inflammatory Agent

CDs have great potential as active pharmaceutical ingredients against various diseases with few side effects. For example, Bridion^®^ is a γ-CD derivative and is in clinical use as a reversal agent for a nondepolarizing neuromuscular blocking agent, rocronium bromide, and vecuronium bromide in clinical use [100]. Clinical trials of HP-β-CD for treatment of Niemann-pick disease type C are underway [101]. In addition, Arima et al. reported that folate-appended methyl-β-CD (FA-M-β-CD) has the potential as anticancer drugs for FR-overexpressing cancer cells through mitophagy [102]. On the other hand, regarding CDEs, Motoyama et al. examined α-CDE (G3, DSC1.9, 3.9, and 4.7) on nitric oxide production in murine macrophages J774.1 cells stimulated with Toll-like receptors (TLR) ligands. α-CDEs (G3, DSC3.9), PAMAM dendrimer (G3), and physical mixture of PAMAM dendrimer (G3) and α-CD significantly inhibited nitric oxide production from J774.1 cells stimulated with TLR ligands [100] in a concentration-dependent manner without cytotoxicity. However, α-CD molecule alone had no effect on nitric oxide production. Hence, the inhibitory effect of α-CDE (G3, DSC3.9) on nitric oxide production might be attributed to PAMAM dendrimer (G3). In addition, increasing the DS value of α-CD in the α-CDE (G3) molecule was accompanied by a significant decrease in the inhibition of nitric oxide production. Meanwhile, the higher gene transfection efficiency of α-CDE (G3)/pDNA complex increased in the order of α-CDE (G3, 1.9) < α-CDE (G3, 3.9) < α-CDE (G3, 4.7). In conclusion, both PAMAM dendrimer (G3) and α-CDEs (G3, DSC1.9, and 3.9) showed anti-inflammatory effects, but α-CDE (G3, DSC4.7) may be considered as a safe gene transfer carrier that does not adversely affect nitric oxide production from macrophages stimulated with TLR ligands [103].

### 13.2. CDE for Treatment of Amyloidosis

As described before, ATTR amyloidosis is caused by the formation of ATTR amyloid fibrils. As ATTR misfolding triggers the formation of aggregates and amyloid fibrils, which are considered to deposit on the tissues, novel clinically effective therapeutic strategies targeted to those processes are urgently needed. Inoue et al. reported that PAMAM dendrimer (G2) inhibited ATTR V30M amyloid fibril formation and reduced already formed ATTR V30M amyloid fibrils by reducing the β-sheet structure of ATTR V30M protein [104]. Additionally, intravenous injection of PAMAM dendrimer (G2) reduced TTR deposition in human ATTR V30M transgenic rats. These results indicate that PAMAM dendrimer (G2) may possess both inhibitory and breaking effects on ATTR V30M amyloid, suggesting that PAMAM dendrimer has the potential as a dual effective agent against TTR amyloidosis [43]. In addition, GUG-β-CDE (G2, DS1.8) was found to inhibit TTR mRNA levels in liver after intravenous injection of its complex with shRNA in TTR V30M transgenic rats. These results suggest that GUG-β-CDE (G2, DS1.8) has a potential for shRNA carrier.

## 14. Conclusions

It is the 20-year anniversary of the first paper on CDE. Arima et al. demonstrated that various CDEs with targeting ligands have been designed for pDNA and oligonucleotide delivery. In addition, CDEs can do co-delivery of pDNA, oligonucleotides, and low-molecular-weight drugs. Additionally, CDEs can form ternary complexes with a low molecular weight of sacran for efficient delivery of pDNA and oligonucleotides. Moreover, PEG-CDEs can form PPRX and PRX with α-CD and/or γ-CD to sustain the release of the pDNA and oligonucleotides from the supramolecules. Interestingly, CDE themselves have the potential as active pharmaceutical ingredients. Most recently, CDE is reported to be a useful Cas9 RNP carrier that induces genome editing in the brain. However, the detailed mechanism by which CDEs improve pDNA, oligonucleotides, and the other drugs, and lower cytotoxicity of PAMAM dendrimers is still unknown. Additionally, further higher functions such as stimuli-responsive systems and biodegradable properties of CDEs from the safety point of view, are needed. Multifunctional CDEs, i.e., those that possess novel potential as oligonucleotide carriers, anti-amyloid agents, anti-inflammatory drugs and imaging agents, etc. for the treatment of Alzheimer’s disease are expected to be applied to pharmaceuticals in the future.

## Figures and Tables

**Figure 1 pharmaceutics-13-00697-f001:**
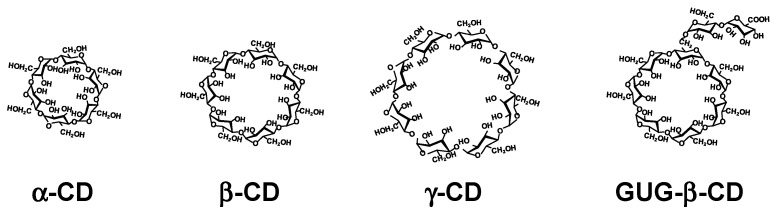
Chemical structure of α-CD, β-CD, γ-CD and glucuronylglucosyl-β-cyclodextrin (GUG-β-CD).

**Figure 2 pharmaceutics-13-00697-f002:**
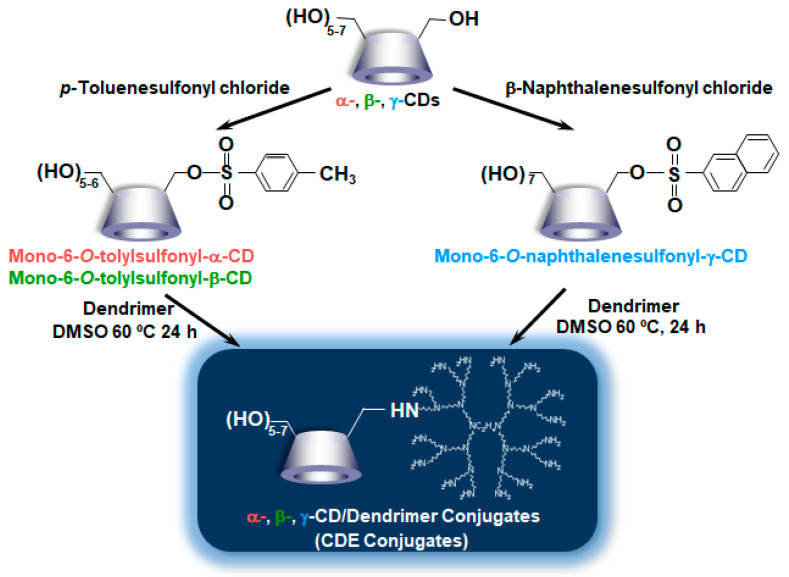
Scheme for preparation of α-CDE (G2, DSC1), β-CDE (G2, DSC1), and γ-CDE (G2, DSC1).

**Figure 3 pharmaceutics-13-00697-f003:**
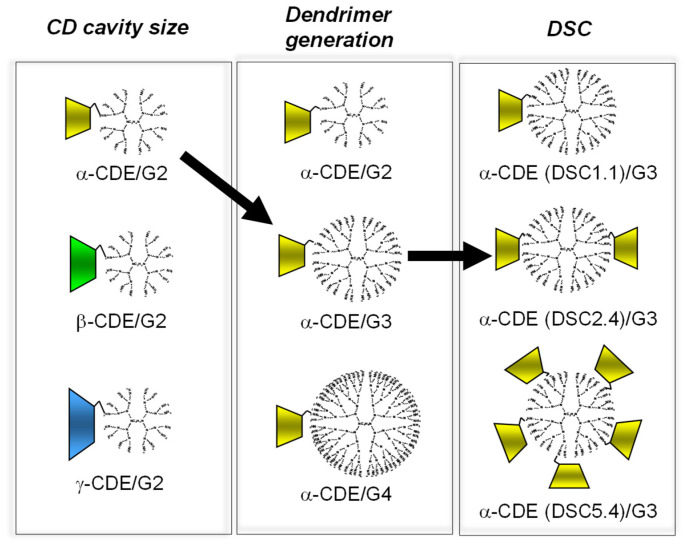
Optimization scheme of α-CDEs as a pDNA carrier.

**Figure 4 pharmaceutics-13-00697-f004:**
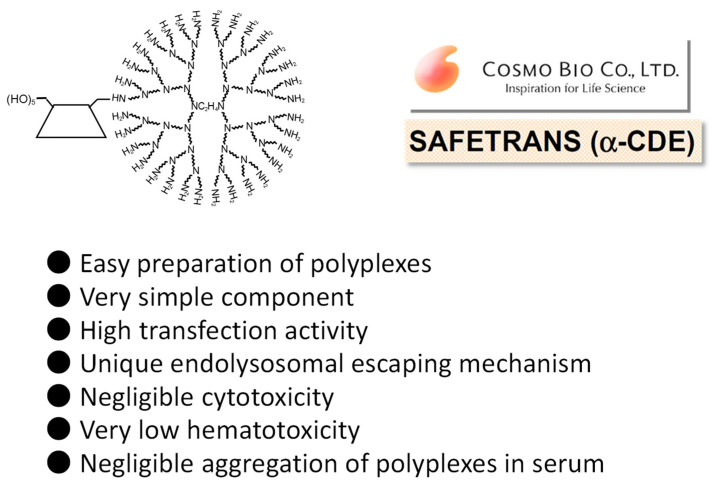
Excellent properties of α-CDE (G3) as a carrier of pDNA, oligonucleotides, and other drugs.

**Figure 5 pharmaceutics-13-00697-f005:**
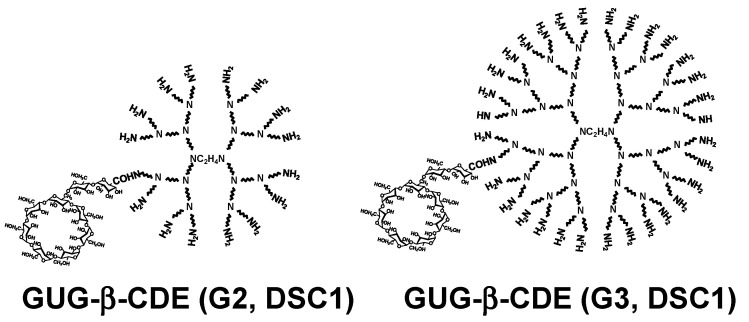
Chemical structure of GUG-β-CDEs.

**Figure 6 pharmaceutics-13-00697-f006:**
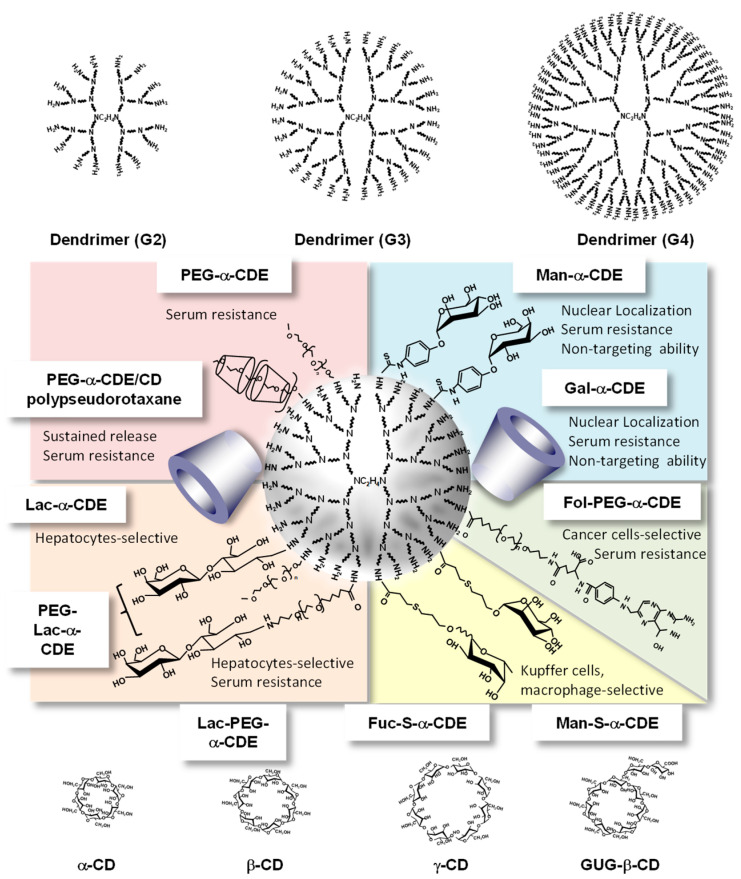
Chemical structures and properties of representative ligand-appended CDEs described in this review.

**Figure 7 pharmaceutics-13-00697-f007:**
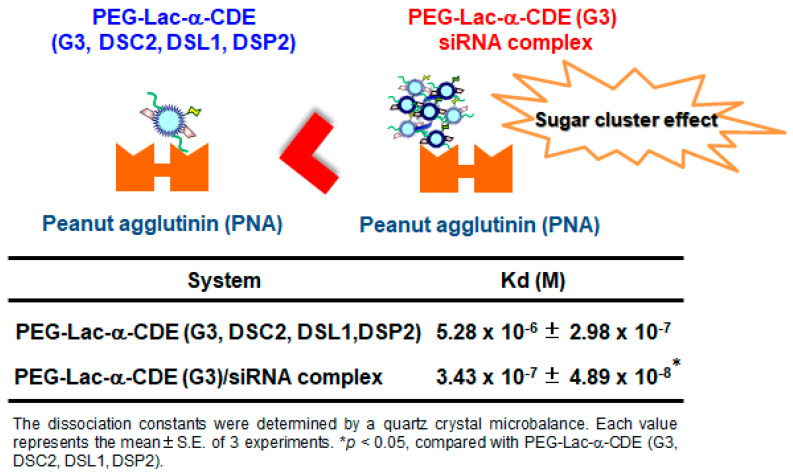
Interaction of PEG-Lac-α-CDE (G3, DSC2, DSL1, DEP2) and its siRNA polyplex with Peanut agglutinin. The dissociation constants were determined by a quartz crystal microbalance. Each value represents the mean ±S.E. of 3 experiments. * *p* < 0.05, compared with PEG-Lac-α-CDE (G3, DSC2, DSL1, DSP2).

**Figure 8 pharmaceutics-13-00697-f008:**
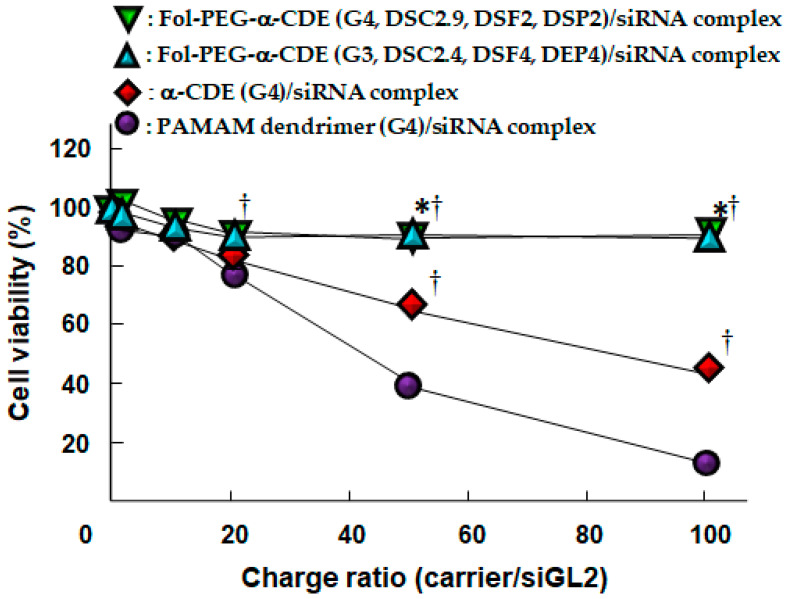
Cytotoxicity of various carrier/siRNA complex in KB cells (FR (+)). The cells were incubated with carrier/siRNA complexes for 24 h. Cell viability was assayed by WST-1 method. The concentration of siGL2 was 100 nM. Culture medium was supplemented with 10% FBS. Each point represents the mean ± S.E. of 3–4 experiments. * *p* < 0.05, compared with α-CDE (G4). ^†^
*p* < 0.05, compared with dendrimer (G4)/siRNA complex.

**Figure 9 pharmaceutics-13-00697-f009:**
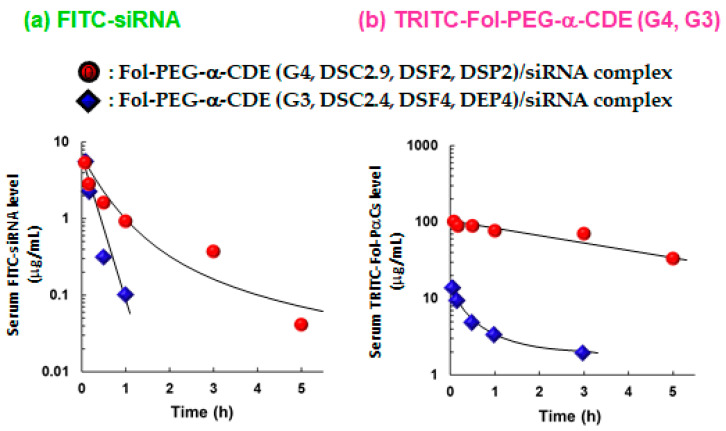
Time courses of blood levels of (**a**) FITC-siRNA and (**b**) TRITC-Fol-PEG-α-CDE (G4, DSC2.9, DSF2, DSP2) and TRITC-Fol-PEG-α-CDE (G3, DSC2.4, DSF4, DEP4) after intravenous injection of a solution containing TRITC-Fol-PEG-α-CDE (G4, DSC2.9, DSF2, DSP2)/FITC-siRNA complex and TRITC-Fol-PEG-α-CDE (G3, DSC2.4, DSF4, DEP4)/FITC-siRNA complex to tail vein of BALB/c mice bearing Colon-26 tumor cells. The charge (N/P) ratio of TRITC-Fol-PEG-α-CDE (G4, DSF2)/FITC-siRNA was 10. The amount of siRNA was 10 mg. Each point represents the mean±S.E. of 3–4 experiments.

**Figure 10 pharmaceutics-13-00697-f010:**
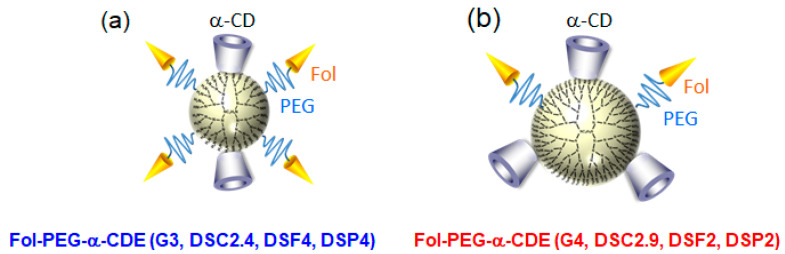
Proposed chemical structures of (**a**) Fol-PEG-α-CDE (G3, DSC2.4, DSF4, DEP4) and (**b**) Fol-PEG-α-CDE (G4, DSC2.9, DSF2, DEP2) as siRNA carriers.

**Figure 11 pharmaceutics-13-00697-f011:**
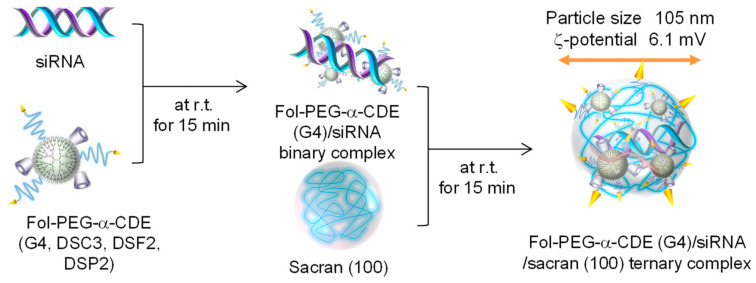
Preparation protocol and properties of Fol-PEG-α-CDE (G4, DSC2.9, DSF2, DEP2)/siRNA/sacran 100) ternary complex.

**Figure 12 pharmaceutics-13-00697-f012:**
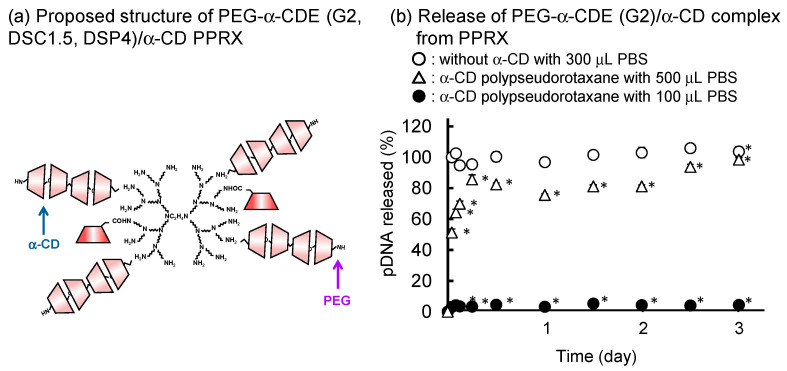
(**a**) Proposed structure of PEG-α-CDE (G2, DSC1.5, DSP4)/α-CD PPRX and (**b**) in vitro release profiles of pDNA from PEG-α-CDE/pDNA/CDs polypseudorotaxanes in different volumes of phosphate-buffered saline (pH 7.4). * *p* < 0.05.

**Table 1 pharmaceutics-13-00697-t001:** Structure and properties of parent cyclodextrins (CD) and glucuronylglucosyl-β-cyclodextrin (GUG-β-CD), based on References [10,17,18].

CD	Glucose Unit	Molecular Weight	Cavity Size (Å)	Cavity Volume (Å ^2^)	Solubility (g/100 mL) ^1^	Surface Tension (mN/m)
α-CD	6	973	4.7–5.3	174	14.5	73
β-CD	7	1135	6.0–6.5	262	1.85	73
γ-CD	8	1297	7.5–8.3	427	23.2	73
GUG-β-CD	9	1473	6.0–6.5	262	>200	73

^1^ In water at 25 °C. ^2^ In water at 20 °C.

**Table 2 pharmaceutics-13-00697-t002:** Comparison of α-CDE and GUG-β-CDE as carriers for gene, oligonucleotides, and other drugs.

Pamameter	Comparison
Physicochemical properties	
Complexation ability	α-CDE > GUG-β-CDE
Particle sizes	α-CDE ≈ GUG-β-CDE
ζ-Potential	α-CDE ≈ GUG-β-CDE
Stability against nuclease	α-CDE ≈ GUG-β-CDE
Cellular association	α-CDE ≈ GUG-β-CDE
Endocytosis pathway	α-CDE ≈ GUG-β-CDE
Endolysosomal escaping ability	α-CDE < GUG-β-CDE
Nuclear localization ability	α-CDE < GUG-β-CDE
Decompaction ability	α-CDE < GUG-β-CDE

**Table 3 pharmaceutics-13-00697-t003:** Comparison of Fol-PEG-α-CDE (G3, DSC2.4, DSF4, DEP4) and Fol-PEG-α-CDE (G4, DSC2.9, DSF2, DEP2) as siRNA carriers.

Properties	Fol-PEG-α-CDE (G3, DSC2.4, DSF4, DSP4)	Fol-PEG-α-CDE (G4, DSC2.9, DSF2, DSP2)
Optimal charge (N/P) ratio	20	10
RNAi effect in the presence of serum	up to 10% FBS	up to 50% FBS
Complexation ability with siRNA	+	++
In vitro RNAi effect	50–60%	50–70%
Cellular uptake	+	++
FR-α selectivity	+++	+++
Interferon response	−	−

−, negligible effect; +, slight effect; ++, moderate effect; +++, strong effect.

**Table 4 pharmaceutics-13-00697-t004:** Various multifunctional CDEs introduced in the review.

CDE	G ^1^	DSC ^2^	DSL ^3^	DSP ^4^	Payload	System	Reference
α-CDE	2	1, 1.2	-	-	pDNA	Complex	[37,38,45,46,47,52,62,63,64,65]
α-CDE	3	1, 1.1, 5.4	-	-	pDNA	Complex	[38,39]
α-CDE	3	2.4	-	-	pDNA	Complex	[38,39,42,44,53,78]
α-CDE	3	2.4	-	-	shRNA	Complex	[42]
α-CDE	3	2.4	-	-	siRNA	Complex	[42,43,78]
α-CDE	3	2.4	-	-	L-HIPRO	Complex	[44]
α-CDE	4	1	-	-	pDNA	Complex	[38]
β-CDE	2	1, 1.3	-	-	pDNA	Complex	[37,45,46,47]
γ-CDE	4	1	-	-	pDNA	Complex	[37]
PEG ^5^-α-CDE	2	1.5	-	4	pDNA	PPRX ^5^	[92]
PEG-α-CDE	2	1	-	3	siRNA	PPRX ^5^	[93]
Gal ^6^-α-CDE	2	1	4	-	pDNA	PPRX ^5^	[51]
Lac ^7^-α-CDE	2	1	1.2, 2.6, 4.6, 6.2, 10.2	-	pDNA	Complex	[52]
Lac-α-CDE	2	2	1	-	pDNA	ternary complex with sacran	[87]
Lac-α-CDE	3	2.4	1.2, 2.6, 4.1, 6.1	-	pDNA	Complex	[53]
Lac-α-CDE	3	2.4	1.2	-	siRNA	Complex	[54,55]
PEG-Lac-α-CDE	3	2.0	1.2	2.1, 4.0, 6.2	pDNA	Complex	[56]
PEG-Lac-α-CDE	3	2.0	1	2	pDNA	Complex	[57]
Man ^8^-α-CDE	2	1	1, 3, 5		pDNA	Complex	[62]
Man-α-CDE	2	1.1	1.1, 3.3, 4.9, 8.3		pDNA	Complex	[63]
Man-α-CDE	3	2.2	5, 10, 13, 20		pDNA	Complex	[64,65]
Man-S-α-CDE	3	2	4		siRNA	Complex	[67,69]
Fuc ^9^-S-α-CDE	2	1	2		decoy DNA	Complex	[72,73]
Fol ^10^-α-CDE	3	2.4	2, 5, 7		pDNA	Complex	[78]
Fol-PEG-α-CDE	3	2.4	2, 5, 7	2,5,7	pDNA	Complex	[78]
Fol-PEG-α-CDE	3	2	2, 4, 7	2, 4, 7	siRNA	Complex	[80]
Fol-PEG-α-CDE	4	2.9	2	2	siRNA	Complex	[81]
Fol-PEG-α-CDE	4	3	2	2	siRNA	ternary complex with sacran	[88]
GUG ^11^-β-CDE	2	1.8			pDNA	Complex	[45,46,47,48]
GUG-β-CDE	2	1.8, 2.5, 3.0, 5.0			siRNA	Complex	[48]
GUG-β-CDE	3	1.6, 3.0, 3.7, 5.0, 8.6			pDNA	Complex	[49]
GUG-β-CDE	3	1.5, 3, 6.5			Cas9 RNP	Complex	[99]
Fol-PEG-GUG-β-CDE	3	3.7	3.9, 6.7, 7.3	3.9, 6.7, 7.3	siRNA	Complex	[83]
Fol-PEG-GUG-β-CDE	3	3.7	6.7	6.7	siRNA, doxorubicin	ternary complex	[90]

^1^ Generation of PAMAM dendrimer, ^2^ degree of substitution of CD, ^3^ degree of substitution of ligand, ^4^ degree of substitution of PEG, ^5^ polyethylene glycol, ^6^ galactose, ^7^ lactose, ^8^ mannose, ^9^ fucose, ^10^ folate, ^11^ glucuronylglucosyl.

## Data Availability

The data presented in this study are openly available.

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
