# Peer review of "Twenty Years of Research on Cyclodextrin Conjugates with PAMAM Dendrimers"

_pharmaceutics, 2021, doi:10.3390/pharmaceutics13050697_

Round 1
Reviewer 1 Report
The topic of this review is very interesting. It can be accepted after minor amendments.
(1) Any toxicological data?
(2) What are the in vivo biological fates of such excipients?
(3) The drug loading capacity should be compared.
Reviewer 2 Report
The Authors proposed a paper entitled “Twenty years of research on cyclodextrin conjugates with PAMAM Dendrimers” for the publication in Pharmaceutics Journal of MDPI. This paper has been proposed as review.
This paper is very interesting, especially for the full state of the art proposed. Also, a good number of original sketches and pictures supported the topics proposed here.
Since the topic are particularly wide and worth of investigation, I would have expected some references about generic description of drug carriers, i.e. classification, administration, and release profiles.
The use of English is quite good, some parts needs a second reading in order to make some paragraphs more fluent.
I have some minor issues reported here:
Affiliation needs some information more, such as the country of the university.
A double space at the end of Line 7.
Cas9 RNP has not been defined in the Abstract, Line 17.
Line 27. What about United States, Africa and South America?
Table 1 whose caption is “structure and properties of parent CD has the same numeration of the following table, concerning the various multifunctional CDEs introduced in the review. Both tables are named as Table 1. Please modify the numeration.
In the first table 1, the reference are needed, as correctly reported in the following table.
Line 180-184. The sentence here is interrupted by the presence of a figure. Maybe better to move the figure after the end of the sentence.
Line 213. “to next optimize” I suggest using another expression.
Due to the large number of acronyms and abbreviations, I suggest adding an abbreviation list, according to the journal guidelines.
However, there are several acronyms that have not been defined at all. I suggest to revise the paper accordingly from this point of view, taking into account that not all the readers will be medical doctors or biologists.
Line 259. Define RNAi.
Title of paragraph 2.4. what is that symbol before “CDE”?
Line 372. Table 2 should be Table 3.
Line 518. Here we have a figure that contains a sort of table inside. I suggest to give the information of the table directly in the caption of this figure and to modify this figure according to this.
Line756. Here we have the same problem presented above. In this case, maybe, the table is necessary and needs to be indicated as table. Then, the figures could remain as figures 7a and 7b.
Line 978. Could you develop the future application stated in this last sentence?
Thank you.
Reviewer 3 Report
The author presents an overview of the 20 years of development
of cyclodextrin-modified dendrimers and studies of their use as oligonucleotides carriers and other areas, to which he largely attributed.
This summary of results, which will be interesting for researchers working in these fields, deserves to be published. Nevertheless, the author should address the following issues first:
The "ON" abbreviation is usually used for oligonucleotides (including oligodeoxynucleotides). Nevertheless, the author is using ODN for oligonucleotides, the abbreviation that is used mainly only for oligodeoxynucleotides. I suggest the author use the commonly used (not so confusing) abbreviations.
A quite recent review about oligonucleotides drug delivery should be mentioned:
Roberts, T. C.; Langer, R.; Wood, M. J. A. Advances in Oligonucleotide Drug Delivery. Nat. Rev. Drug Discovery 2020, 19 (10), 673–694. https://doi.org/10.1038/s41573-020-0075-7.
The sentences on lines 76,77 are confusing: the difference between "polymers for DDS" and "polymer-based carriers" should be clearly explained. Also, "there are no example" on line 77 (and in the Abstract) was probably meant "there was no example"? (The whole review is about such examples.)
The English should be checked thoroughly so that the sentences' intended meaning is right, e.g., on lines 86, 114, 121, 331, 892 (unfinished sentence), 932, 975 (detailed mechanism?)
Line 84 - If the synthetically prepared CDs with 3 and 4 Glc units were mentioned, references to CDs with 5 or more than 8 Glc units should be given.
Line 94 - The structures in Figure 1 should be fixed: for alpha-, beta- and gamma-CD, the hydroxyls in Glc positions 3 are drawn almost as axial (they should be equatorial). The GUG-beta-CD is shown as its mirror image
(i.e., as the unknown L-glucose-based isomer), in all occurrences in the text.
Line 99 - Table 1, the value of solubility of GUG-beta-CD should be >200 g/100 mL (not >200,000)
Line 159 - This should probably be section 3. (Section 2. is on page 73.)
Line 165 - The term DSC should be explained more thoroughly, "degree of substitution" is used in CD chemistry for describing the number of substituents on CD molecules, but here it defines the number of CD units bound to the dendrimer (as shown later).
Line 180, Figure 2, the names of the arylsulfonyl-CDs are incorrect if the author would follow carbohydrate IUPAC nomenclature. It should be either mono-6-O-arylsulfonyl-CD, or better 6I-O-arylsulfonyl-CD, or 6A-O-arylsulfonyl-CD is also used. If the use of "deoxy" is needed for consistency with other names, then the names should be 6-deoxy-6-(arylsulfonyloxy)-CD.
